# High dimensional, tabular deep learning with an auxiliary knowledge graph

**Camilo Ruiz**[1,2,*], **Hongyu Ren**[1,*], **Kexin Huang**[1], **Jure Leskovec**[1]
[1]Department of Computer Science, Stanford University
[2]Department of Bioengineering, Stanford University
[*]Equal contribution
{caruiz, hyren, kexinh, jure}@cs.stanford.edu

## Abstract

Machine learning models exhibit strong performance on datasets with abundant labeled samples. However, for tabular datasets with extremely high $d$-dimensional features but limited $n$ samples (i.e. $d \gg n$), machine learning models struggle to achieve strong performance due to the risk of overfitting. Here, our key insight is that there is often abundant, auxiliary domain information describing input features which can be structured as a heterogeneous knowledge graph (KG). We propose PLATO, a method that achieves strong performance on tabular data with $d \gg n$ by using an auxiliary KG describing input features to regularize a multilayer perceptron (MLP). In PLATO, each input feature corresponds to a node in the auxiliary KG. In the MLP's first layer, each input feature also corresponds to a weight vector. PLATO is based on the inductive bias that two input features corresponding to similar nodes in the auxiliary KG should have similar weight vectors in the MLP's first layer. PLATO captures this inductive bias by inferring the weight vector for each input feature from its corresponding node in the KG via a trainable message-passing function. Across 6 $d \gg n$ datasets, PLATO outperforms 13 state-of-the-art baselines by up to 10.19%.

## 1 Introduction

Machine learning models have reached state-of-the-art performance in domains with abundant labeled data like computer vision [76, 10] and natural language processing [70, 12, 52]. However, for tabular datasets in which the number $d$ of features vastly exceeds the number $n$ of samples, machine learning models struggle to achieve strong performance [24, 41]. Crucially, many tabular datasets from scientific domains [21, 30, 79, 17, 16, 33] have high-dimensional features but limited labeled samples due to the high time and labor costs of experiments. For these and other tabular datasets with $d \gg n$, the performance of machine learning models is currently limited.

The key challenge for machine learning models when $d \gg n$ is the risk of overfitting. Indeed, deep models can have a large number of trainable weights, yet training is limited by the comparatively small number of labeled samples. As a result, tabular deep learning approaches so far have focused on data-rich regimes with far more samples than features ($n \gg d$) [19, 18, 59]. In the low-data regime with far more features than samples ($d \gg n$), the dominant approaches for single tabular datasets are still statistical methods [24]. These statistical methods reduce the dimensionality of the input space [1, 41, 66, 67], select features [64, 8, 14, 46], impose regularization penalties on parameter magnitudes [45], or use ensembles of weak tree-based models [15, 7, 34, 42, 51].

Here, we present a novel problem setting and framework that enables tabular deep learning when $d \gg n$ (Figure 1). Our key insight is that there is often abundant, auxiliary domain information

37th Conference on Neural Information Processing Systems (NeurIPS 2023).

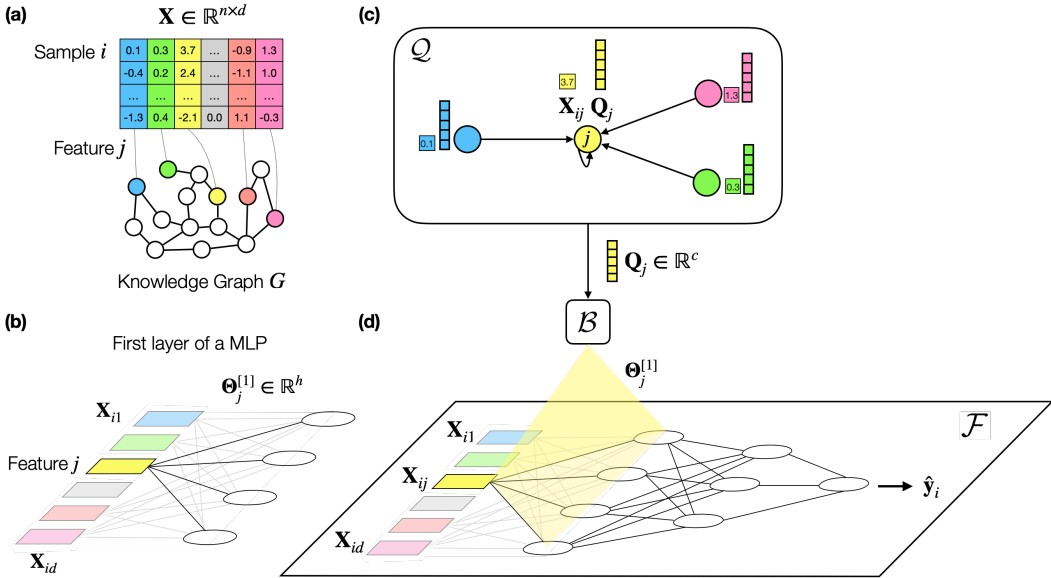

Figure 1: **PLATO is a method that uses auxiliary domain information describing input features to regularize a multilayer perceptron (MLP) and achieve strong performance on tabular data with $d \gg n$.** **(a)** In PLATO, each input feature $j$ corresponds to a node in an auxiliary KG of domain information. **(b)** In the first layer of a MLP with $h$ hidden units, each input feature $j$ corresponds to a vector of weights $\boldsymbol{\Theta}_j^{[1]} \in \mathbb{R}^h$ such that the weight vectors of all $d$ features compose the weight matrix $\boldsymbol{\Theta}^{[1]} \in \mathbb{R}^{d \times h}$. PLATO is based on the inductive bias that, if two input features $j$ and $k$ correspond to similar nodes in the auxiliary KG, they should have similar weight vectors $\boldsymbol{\Theta}_j^{[1]}$ and $\boldsymbol{\Theta}_k^{[1]}$ in the MLP. **(c,d)** PLATO captures this inductive bias by inferring the weight vector for each input feature $j$ from its corresponding node in the KG. A trainable message-passing function $\mathcal{Q}$ creates a low-dimensional embedding $\mathbf{Q}_j \in \mathbb{R}^c$ for each input feature $j$. A neural network $\mathcal{B}$ that is shared across all input features then infers the weight vector $\boldsymbol{\Theta}_j^{[1]}$ corresponding to input feature $j$ from $\mathbf{Q}_j$. Input features with similar embeddings produce similar weight vectors, regularizing the MLP.

describing input features which can be structured as a heterogeneous knowledge graph (KG). We propose a novel problem setting in which each input feature of a tabular dataset corresponds to a node in an auxiliary KG (Figure 1a). To represent diverse domain information describing the input features, the KG contains feature and non-feature nodes as well as multiple node and edge types. For example, consider a tabular medical dataset in which each row is a cancer patient, each column is a gene, and each value is the amount of a gene in the patient's tumor. For this tabular dataset, there exists an auxiliary KG with each gene (*i.e.* input feature) as a node. Each gene node has edges to other gene nodes (*i.e.* other feature nodes) with diverse edge types like "activates" or "inhibits." Each gene node also has edges to other nodes (*i.e.* non-feature nodes) representing the gene's function in the body like "heart rate". Finally, the function nodes (*i.e.* non-feature nodes) have edges to each other representing their anatomical relationships like "heart rate"-"part of"-"cardiac system". Note that the KG does *not* capture the relationships between *input data samples* but instead captures the relationships between *input features* and other domain information.

Within our novel problem setting, we propose PLATO, a method that enables deep learning for tabular data with $d \gg n$ by using an auxiliary KG describing input features (Figure 1). PLATO achieves strong performance by using the auxiliary KG to regularize a multilayer perceptron (MLP). In PLATO, each input feature corresponds to a node in the auxiliary KG (Figure 1a). In the first layer of the MLP, each input feature also corresponds to a weight vector such that the weight vectors of all features collectively compose the weight matrix (Figure 1b). PLATO is based on the inductive bias that two input features which correspond to similar nodes in the KG should have similar weight vectors in the first layer of the MLP. PLATO captures this inductive bias by inferring the weight vector for a feature from its corresponding node in the auxiliary KG with a trainable message-passing function (Figure 1c,d). Inferring the weights in the MLP's first layer also leads to a drastic reduction in the number of trainable weights, since most weights in a MLP are usually in the first layer when $d \gg n$.

We exhibit PLATO's performance on 6 $d \gg n$ tabular datasets with 13 state-of-the-art baselines spanning dimensionality reduction, feature selection, statistical models, graph regularization, weight-inference, and tabular deep learning. Following a rigorous evaluation protocol from the tabular deep learning literature [19, 18], PLATO outperforms the prior state-of-the-art on all 6 datasets by up to 10.19%. Ablation studies demonstrate the importance of PLATO's trainable message-passing, the importance of non-feature nodes in the KG, and PLATO's robustness to missing edges in the KG. Ultimately, PLATO enables deep learning for tabular data with $d \gg n$ by using an auxiliary KG describing the input features.

## 2 Related Work

**Tabular deep learning methods.** In contrast to PLATO's setting, tabular deep learning methods have primarily been developed for settings with far more samples than features (*i.e.* $n \gg d$). Indeed, recent tabular deep learning benchmarks ignore datasets with a large number of features and a small number of samples [19, 18, 59]. In the $n \gg d$ setting, various categories of deep tabular models exist. First, decision tree models like NODE [50] make decision trees differentiable to enable gradient-based optimization [25, 36, 80]. Second, multilayer perceptrons (MLPs) apply sequential, non-linear transformations to input features [32, 31]. Third, tabular transformer architectures use an attention mechanism to select and learn interactions among features. Examples include TabNet [3], TabTransformer [29], FT-Transformer [18], TabPFN [26], SAINT [61], Non-Parametric Transformers [37], and AutoInt [62]. Finally, although PLATO focuses on single tabular datasets, transfer learning architectures can learn across multiple tabular datasets [39, 84, 73]. Ultimately, we compare PLATO to several benchmarked, state-of-the-art models for single, tabular datasets [19, 18, 59].

$d \gg n$ **methods.** For PLATO's setting in which $d \gg n$, various tabular machine learning approaches exist [24]. First, dimensionality reduction techniques like PCA [1] aim to reduce the dimensionality of the input data while preserving as much of the the variance in the data as possible [41, 66, 67]. Second, feature selection approaches select a parsimonious set of features, leading to a smaller feature space. Feature selection approaches include LASSO [64] and its variants [8, 14, 46]. For feature selection with deep learning, Stochastic Gates [77] are among the best performing of many variants [4, 43]. Finally, tree-based models like XGBoost learn ensembles of weak decision trees models to make an overall prediction [15, 7, 34, 51].

**Weight inference.** Using one network to infer the weights of another has been studied extensively [11, 58, 6]. For example, [22] infers the weights in all layers of a sequential model (*i.e.* RNN, LSTM) by using information about the weights' structure. Diet Networks [54] infer weights by hand-crafting prior information about the input features or using random projections. By contrast, PLATO infers the weights in a MLP from prior information describing the input features in an auxiliary KG. PLATO's weight inference uniquely captures the inductive bias that two input features corresponding to similar nodes in a KG should have similar corresponding weight vectors in the first layer of a MLP (Figure 1).

**Graph regularization.** Graph regularization approaches regularize the weights of a linear model based on a simple graph between input features. The graph is typically constructed from the tabular data based on covariance relationships. Approaches then add a regularization penalty to the loss function which forces the weights of the linear model to vary smoothly over the corresponding feature nodes in the graph. State-of-the-art methods include GraphNet [20] and Network-Constrained LASSO [40] which are based on a Laplacian regularization [60, 2] as well as Network LASSO [23] which generalizes the Group LASSO [83] to a network setting. PLATO differs from graph regularization approaches in two key ways. First, PLATO's KG includes both feature and non-feature nodes and multiple edge types, thereby modeling diverse, prior domain information that is missing in graph regularization approaches. Second, PLATO infers the weights of a deep non-linear model (*i.e.* a MLP) rather than adding a regularization penalty to a loss, representing a distinct regularization mechanism.

**Knowledge graph methods.** Existing KG approaches are designed for tasks directly on the graph like link prediction or node classification [71, 65, 72, 78, 13]. By contrast, PLATO does not make any predictions on the KG. Instead, PLATO makes predictions on a separate, tabular dataset by using the KG as a prior. Graph classification methods also do not apply (Appendix B).

# 3 PLATO

PLATO is a machine learning method for tabular datasets with $d \gg n$ and an auxiliary knowledge graph (KG) with input features as nodes (Section 3.1). PLATO's key insight is that there often exists abundant domain information describing input features which can be structured as an auxiliary KG $G$ (Figure 1a). PLATO uses the auxiliary KG to regularize a multilayer perceptron (MLP) and achieve strong performance on tabular data when $d \gg n$.

## 3.1 Problem setting

Consider a tabular dataset $\mathbf{X} \in \mathbb{R}^{n \times d}$ with labels $\mathbf{y} \in \mathbb{R}^n$ and far more $d$ features than $n$ samples such that $d \gg n$. The goal is to train a model $\mathcal{F}$ to predict labels $\hat{\mathbf{y}}$ from the input $\mathbf{X}$. PLATO assumes the existence of an auxiliary knowledge graph $G = (V, E)$ with $|V|$ nodes and $|E|$ edges such that each input feature $j$ corresponds to a node in $G$. Formally, $\forall j \in \{1, \ldots, d\}, \exists v \in V$ s.t. $j \mapsto v$, as shown in Figure 1a. $G$ also contains additional nodes which represent broader knowledge describing the domain. The edges in $G$ are (head node, relation type, tail node) triplets.

## 3.2 PLATO's inductive bias

In PLATO, each input feature $j$ corresponds to a node in the auxiliary KG (Figure 1a). In the first layer of a MLP with $h$ hidden units, each input feature $j$ also corresponds to a weight vector $\Theta_j^{[1]} \in \mathbb{R}^h$ such that the weight vectors of all features collectively compose the weight matrix $\Theta^{[1]} \in \mathbb{R}^{d \times h}$ (Figure 1b). PLATO is based on the inductive bias that two input features $j$ and $k$ which correspond to similar nodes in the KG should have similar weight vectors $\Theta_j^{[1]}$ and $\Theta_k^{[1]}$ in the first layer of the MLP. PLATO captures this inductive bias by inferring the weight vector for a feature from its corresponding node in the auxiliary KG with a trainable message-passing function (Figure 1c,d).

## 3.3 PLATO overview

PLATO has four key steps. First, PLATO uses a self-supervised objective on the auxiliary KG to pretrain an embedding for each input feature (Section 3.4). Second, PLATO updates each feature embedding with a trainable message-passing function that is trained on the supervised loss objective for the tabular data (Section 3.5, Figure 1c). Third, PLATO infers the weights in the first layer of the MLP from the feature embeddings with a small neural network that is shared across input features (Section 3.6, Figure 1d). Finally, the MLP predicts the label for the input sample.

## 3.4 Pretraining feature embeddings with self-supervision on the knowledge graph

First, PLATO learns general prior information about each input feature $j$ from the auxiliary KG $G$. PLATO represents the general prior information about each input feature $j$ as a low-dimensional embedding $\mathbf{M}_j \in \mathbb{R}^c$. Since each input feature $j$ corresponds to a node in $G$, PLATO can learn $\mathbf{M}_j$ by learning an embedding for the corresponding feature node in $G$. Any self-supervised node embedding method on $G$ can be used within PLATO's framework.

**Formal notation.** Formally, PLATO uses self-supervision on $G$ to pretrain an embedding for each input feature according to

$$\mathbf{M} = \mathcal{H}(G). \tag{1}$$

$\mathbf{M} \in \mathbb{R}^{d \times c}$ is the matrix of all feature embeddings. $\mathcal{H}$ is a self-supervised node embedding method. We refer to Eq. (1) as pretraining since only the auxiliary KG $G$ is used but the tabular data $\mathbf{X}, \mathbf{y}$ is ignored. After pretraining, the feature embeddings $\mathbf{M}$ are fixed.

For $\mathcal{H}$, we choose ComplEx as it is a prominent and highly scalable KG node embedding method [65]. ComplEx uses a self-supervised objective which learns an embedding for each node in $G$ by classifying whether a proposed edge exists in $G$. ComplEx's proposed edges include both feature nodes and other nodes in $G$, thereby integrating prior information about the input features and the broader domain. We also test KG embedding methods DistMult [78] and TransE [72] in Appendix C.

## 3.5 Updating feature embeddings with a message-passing function trained on tabular data

PLATO next updates each feature embedding with a trainable message-passing function that is trained on the supervised loss for the tabular data (Figure 1c). During message-passing, PLATO updates the embedding of each input feature to be a weighted aggregation of it's neighbors' embeddings.

**Formal notation.** Formally, PLATO uses a message-passing function $\mathcal{Q}$ on the KG to update each pre-trained feature embedding $\mathbf{M}_j \in \mathbb{R}^c$ to feature embedding $\mathbf{Q}_j \in \mathbb{R}^c$ according to

$$\mathbf{Q} = \mathcal{Q}(\mathbf{M}, G, \mathbf{X}_i; \mathbf{\Pi}). \tag{2}$$

As input, the message-passing function considers the pre-trained feature embeddings $\mathbf{M}$, the knowledge graph $G$, and the sample value $\mathbf{X}_i$. $\mathcal{Q}$ uses an attention mechanism which considers the sample value $\mathbf{X}_i$. The only trainable weights in $\mathcal{Q}$ are in the attention mechanism and are $\mathbf{\Pi}$.

**The message passing network $\mathcal{Q}$.** Let $\mathbf{Q}_j^{[r]}$ be the embedding of input feature $j$ after round $r \in \{1, ..., R\}$ of message passing. For each input feature $j$, $\mathcal{Q}$ first initializes the updated feature embedding to the pretrained feature embedding.

$$\mathbf{Q}_j^{[0]} = \mathbf{M}_j. \tag{2a}$$

$\mathcal{Q}$ then conducts $R$ rounds of message passing. In each round of message passing, the feature embedding $\mathbf{Q}_j^{[r]}$ is updated from the feature embedding of each neighbor $k$ in the prior round $\mathbf{Q}_k^{[r-1]}$ and its own feature embedding in the prior round $\mathbf{Q}_j^{[r-1]}$. The "message" being passed is the embedding of each feature from the prior round.

$$\mathbf{Q}_j^{[r]} = \sigma \left[ \overbrace{\beta \left( \sum_{k \in N_j} \alpha_{ijk} \mathbf{Q}_k^{[r-1]} \right)}^{\text{Weighted messages from neighbors}} + \underbrace{(1 - \beta) \mathbf{Q}_j^{[r-1]}}_{\text{Weighted message from self}} \right]. \tag{2b}$$

$\sigma$ is an optional nonlinearity. $N_j$ are the neighbors of feature node $j$ in $G$.

During message-passing, $\mathcal{Q}$ uses two scalar values $\beta \in \mathbb{R}$ and $\alpha_{ijk} \in \mathbb{R}$ to control the weights of messages. First, $\mathcal{Q}$ uses hyperparameter $\beta \in \mathbb{R}$ to control the weight of the messages aggregated from the feature node's neighbors vs. from the feature node itself. Second, $\mathcal{Q}$ calculates an attention coefficient $\alpha_{ijk} \in \mathbb{R}$ to allow distinct nodes in the same neighborhood to have distinct weights. The coefficient $\alpha_{ijk}$ specifies the weight of the message between feature $j$ and neighbor $k$ for sample $i$.

After $R$ rounds of message-passing, the updated feature embeddings $\mathbf{Q}_j$ are set.

$$\mathbf{Q}_j = \mathbf{Q}_j^{[R]}. \tag{2c}$$

**The attention coefficient.** PLATO's attention coefficient $\alpha_{ijk}$ is inspired by [68] in which node attributes are used to calculate the weight of a message between neighboring nodes. For a sample $i$ in PLATO, the node attributes for features $j$ and $k$ are their sample values $\mathbf{X}_{ij} \in \mathbb{R}$ and $\mathbf{X}_{ik} \in \mathbb{R}$. PLATO thus uses the sample values $\mathbf{X}_{ij}$ and $\mathbf{X}_{ik}$ to calculate the attention coefficient. The attention coefficient $e_{ijk}$ indicates the importance of node $j$ to node $k$ for sample $i$.

$$e_{ijk} = \mathcal{A}(\mathbf{X}_{ij}, \mathbf{X}_{ik}; \mathbf{\Pi}). \tag{2d}$$

$\mathcal{A}$ is a shallow neural network parameterized by $\mathbf{\Pi}$ that is shared across samples and features. The number of trainable weights in $\mathbf{\Pi}$ is small since the input of $\mathcal{A}$ is $\mathbb{R}^2$ and the output of $\mathcal{A}$ is a scalar $\mathbb{R}$.

To make the attention coefficients comparable across different nodes, PLATO normalizes the attention coefficients with a softmax function across the neighbors $N_j$ of node $j$.

$$\alpha_{ijk} = \text{softmax}_k(e_{ijk}) = \frac{\exp(e_{ijk})}{\sum_{t \in N_j} \exp(e_{ijt})}. \tag{2e}$$

**Algorithm 1:** The PLATO Algorithm.

**Input:** A data sample $\mathbf{X}_i \in \mathbb{R}^d$, a knowledge graph $G$ containing each input feature in $\mathbf{X}$ as a node, a matrix of input feature embeddings $\mathbf{M} \in \mathbb{R}^{d \times c}$ pre-trained over $G$.
**Output:** A predicted label $\hat{\mathbf{y}}_i \in \mathbb{R}$.

1 Use a trainable message-passing function $\mathcal{Q}$ to update the pre-trained feature embeddings:
$\mathbf{Q} = \mathcal{Q}(\mathbf{M}, G, \mathbf{X}_i; \mathbf{\Pi}), \mathbf{Q}_j \in \mathbb{R}^c, \mathbf{Q} \in \mathbb{R}^{d \times c}$
2 Infer the weight vector in the first layer of a MLP that corresponds to an input feature $j$ with a neural network $\mathcal{B}$:
$\hat{\mathbf{\Theta}}_j^{[1]} = \mathcal{B}(\mathbf{Q}_j|\mathbf{X}_i; \mathbf{\Phi}), \hat{\mathbf{\Theta}}_j^{[1]} \in \mathbb{R}^h$
3 Repeat to infer the weight vectors corresponding to all input features by sharing the neural network $\mathcal{B}$:
$\hat{\mathbf{\Theta}}^{[1]} \in \mathbb{R}^{d \times h}$
4 Concatenate the first layer inferred weights with the trainable weights in the rest of the MLP layers:
$\hat{\mathbf{\Theta}} = \{\hat{\mathbf{\Theta}}^{[1]}|\mathbf{X}_i\} \cup \{\mathbf{\Theta}^{[2]}, \dots, \mathbf{\Theta}^{[L]}\}$.
5 Predict the label with a MLP $\mathcal{F}$ that is parameterized by $\hat{\mathbf{\Theta}}$
$\hat{\mathbf{y}}_i = \mathcal{F}(\mathbf{X}_i; \hat{\mathbf{\Theta}}|\mathbf{X}_i), \hat{\mathbf{y}}_i \in \mathbb{R}$
Trainable weights: $\mathbf{\Pi}, \mathbf{\Phi}, \mathbf{\Theta}^{[2]}, \dots, \mathbf{\Theta}^{[L]}$.

## 3.6 Inferring the first layer of weights in $\mathcal{F}$ from the updated feature embeddings

Finally, PLATO infers the weights in the first layer of a MLP $\mathcal{F}$ from the updated feature embeddings (Figure 1d). In the first layer of a MLP with $h$ hidden units, each input feature $j$ corresponds to a weight vector $\mathbf{\Theta}_j^{[1]} \in \mathbb{R}^h$ (Figure 1b). The weight matrix in the first layer of the MLP, $\mathbf{\Theta}^{[1]} \in \mathbb{R}^{d \times h}$, is simply the concatenation of $d$ weight vectors, one corresponding to each input feature. For each input feature $j$, PLATO infers the weight vector $\hat{\mathbf{\Theta}}_j^{[1]} \in \mathbb{R}^h$ from the feature embedding $\mathbf{Q}_j \in \mathbb{R}^c$ by using a shallow neural network shared across input features. Input features with similar feature embeddings will produce similar weight vectors. Thus, PLATO captures the inductive bias that input features corresponding to similar nodes in the KG should have similar corresponding weight vectors in the MLP's first layer.

**Formal notation.** PLATO infers the weight vector associated with each input feature $j$ in the first layer of $\mathcal{F}$ with

$$\hat{\mathbf{\Theta}}_j^{[1]} = \mathcal{B}(\mathbf{Q}_j|\mathbf{X}_i; \mathbf{\Phi}). \tag{3}$$

$\mathcal{B}$ is a shallow neural network with trainable weights $\mathbf{\Phi}$. $\mathbf{Q}_j$ is the updated feature embedding of $j$ which is conditioned on the specific input sample $\mathbf{X}_i$ since the input sample is used as an input in its calculation (Section 3.5, Equation 2). $\mathbf{\Phi}$ are the weights of $\mathcal{B}$. $\mathcal{B}$ and its weights $\mathbf{\Phi}$ are shared across each feature $j \in \{1, \dots, d\}$.

**PLATO drastically reduces the number of trainable weights compared to a standard MLP.** The sharing of $\mathcal{B}$ and $\mathbf{\Phi}$ across all input features drastically reduces the number of trainable weights compared to a standard MLP. For a high-dimensional tabular dataset (*i.e.* $d \gg n$), a standard MLP $\mathcal{T}$ with $h$ hidden units has a large number of trainable weights in the first layer since $\mathbf{\Theta}^{[1]} \in \mathbb{R}^{d \times h}$. A standard MLP $\mathcal{T}$ must learn all $dh$ of these trainable weights by backpropagation. By contrast, $\mathcal{B}$ uses a shared set of trainable weights $\mathbf{\Phi}$ to infer $\hat{\mathbf{\Theta}}_j$ from $\mathbf{Q}_j$ for every $j \in \{1, \dots, d\}$. The number of trainable weights in $\Phi$ is small compared to $dh$ since $\mathcal{B}$ need only transform every $\mathbf{Q}_j \in \mathbb{R}^c$ to $\hat{\mathbf{\Theta}}^{[1]} \in \mathbb{R}^h$. Thus, $|\Phi| = ch$ (assuming $\mathcal{B}$ is a single layer neural network). $c$, the dimensionality of the feature embedding, is much less than $d$ the number of input features. As a result, $|\Phi| = ch \ll dh$ and PLATO drastically reduces the number of trainable weights in the first layer of a MLP.

## 3.7 The PLATO algorithm

PLATO is outlined in Algorithm 1.

Table 1: **PLATO outperforms statistical and deep baselines when $d \gg n$.** For every dataset, the best overall model is in **bold** and the second best model is underlined.

| Dataset | | MNSCLC | CM | PDAC | BRCA | CRC | CH |
|---|---|---|---|---|---|---|---|
| # of features $d$ | | 15,390 | 13,183 | 12,932 | 12,693 | 18,206 | 19,902 |
| # of samples $n$ | | 295 | 286 | 321 | 476 | 562 | 924 |
| $d/n$ | | 52.2 | 46.1 | 40.3 | 28.2 | 22.6 | 19.7 |
| Classic Stat ML | Ridge | $0.153_{\pm0.000}$ | $0.390_{\pm0.000}$ | $0.344_{\pm0.000}$ | $\underline{0.538}_{\pm0.000}$ | $0.376_{\pm0.000}$ | $0.546_{\pm0.000}$ |
| Dim. Reduct. | PCA | $0.156_{\pm0.113}$ | $0.070_{\pm0.000}$ | $0.232_{\pm0.121}$ | $0.452_{\pm0.000}$ | $0.193_{\pm0.163}$ | $0.237_{\pm0.232}$ |
| Feat. Select. | LASSO | $0.168_{\pm0.000}$ | $\underline{0.431}_{\pm0.000}$ | $0.346_{\pm0.000}$ | $0.470_{\pm0.000}$ | $\underline{0.400}_{\pm0.000}$ | $0.547_{\pm0.000}$ |
| | STG | $0.132_{\pm0.130}$ | $0.366_{\pm0.043}$ | $0.258_{\pm0.055}$ | $0.485_{\pm0.037}$ | $0.301_{\pm0.010}$ | $0.262_{\pm0.076}$ |
| Decision Tree | XGBoost | $-0.02_{\pm0.000}$ | $0.225_{\pm0.000}$ | $\underline{0.363}_{\pm0.000}$ | $0.347_{\pm0.000}$ | $0.354_{\pm0.000}$ | $\underline{0.728}_{\pm0.000}$ |
| Graph Reg. | GraphNet | $0.169_{\pm0.030}$ | $0.277_{\pm0.099}$ | $0.249_{\pm0.018}$ | $0.350_{\pm0.069}$ | $0.125_{\pm0.061}$ | $0.646_{\pm0.051}$ |
| | NC LASSO | $0.210_{\pm0.014}$ | $0.339_{\pm0.044}$ | $0.327_{\pm0.053}$ | $0.458_{\pm0.083}$ | $0.220_{\pm0.030}$ | $0.415_{\pm0.083}$ |
| | Network LASSO | $0.212_{\pm0.046}$ | $0.243_{\pm0.058}$ | $0.136_{\pm0.027}$ | $0.348_{\pm0.033}$ | $0.171_{\pm0.040}$ | $0.212_{\pm0.091}$ |
| Param. Infer. | Diet | $-0.04_{\pm0.205}$ | $0.054_{\pm0.149}$ | $0.309_{\pm0.096}$ | $0.213_{\pm0.036}$ | $0.087_{\pm0.112}$ | $0.148_{\pm0.008}$ |
| Tabular DL | MLP | $0.128_{\pm0.126}$ | $0.322_{\pm0.043}$ | $0.289_{\pm0.047}$ | $0.240_{\pm0.067}$ | $0.355_{\pm0.022}$ | $0.044_{\pm0.039}$ |
| | NODE | $0.003_{\pm0.000}$ | $0.150_{\pm0.000}$ | $0.190_{\pm0.000}$ | $0.512_{\pm0.000}$ | $0.344_{\pm0.000}$ | $0.181_{\pm0.000}$ |
| | TabTransformer | $\underline{0.265}_{\pm0.000}$ | $0.072_{\pm0.000}$ | $0.029_{\pm0.000}$ | $0.202_{\pm0.000}$ | $0.238_{\pm0.000}$ | $0.020_{\pm0.000}$ |
| | TabNet | $0.085_{\pm0.028}$ | $0.010_{\pm0.068}$ | $0.088_{\pm0.037}$ | $0.055_{\pm0.037}$ | $0.018_{\pm0.016}$ | $0.039_{\pm0.026}$ |
| Ours | PLATO | $\mathbf{0.272}_{\pm\mathbf{0.130}}$ | $\mathbf{0.435}_{\pm\mathbf{0.022}}$ | $\mathbf{0.400}_{\pm\mathbf{0.021}}$ | $\mathbf{0.583}_{\pm\mathbf{0.019}}$ | $\mathbf{0.401}_{\pm\mathbf{0.019}}$ | $\mathbf{0.770}_{\pm\mathbf{0.003}}$ |

Table 2: **PLATO's performance depends on updating feature embeddings with a trainable message-passing (MP) function.**

| Weight Infer. $\mathcal{B}$ Input | Feature Info. | Trainable MP | PearsonR |
|---|---|---|---|
| Updated feat. embed. $\mathbf{Q}$ | ✔ | ✔ | $0.583_{\pm0.019}$ |
| General feat. embed $\mathbf{M}$ | ✔ | ✘ | $0.522_{\pm0.030}$ |
| None | ✘ | ✘ | $0.240_{\pm0.067}$ |

Table 3: **PLATO's performance depends on both feature nodes in $G$ and other nodes representing broader domain information.**

| Auxiliary KG | Feature Info. | Broader Info. | PearsonR |
|---|---|---|---|
| Full KG | ✔ | ✔ | $0.583_{\pm0.019}$ |
| Feature-only KG | ✔ | ✘ | $0.539_{\pm0.038}$ |
| No KG | ✘ | ✘ | $0.240_{\pm0.067}$ |

## 4 Experiments

We evaluate PLATO against 13 baselines on 10 tabular datasets (6 with $d \gg n$, 4 with $d \sim n$).

**Datasets.** We use 6 tabular $d \gg n$ datasets, 4 $d \sim n$ datasets [16, 17, 30, 79], and a KG from prior studies [44, 35, 38, 56, 63, 74, 75] (Appendix G, H). The KG contains 108,447 nodes, 3,066,156 edges, and 99 relation types. All datasets include features which map to a subset of knowledge graph nodes. Code, data, and the KG are available at `https://github.com/snap-stanford/plato`.

**Baselines.** We compare PLATO to 13 state-of-the art statistical and deep baselines. We consider regularization with Ridge Regression [45], dimensionality reduction with PCA [1] followed by linear regression, feature selection with LASSO [64], deep feature selection with Stochastic Gates [77], and gradient boosted decision trees with XGBoost [7]. We consider tabular deep learning with a standard MLP, self-attention-based methods with TabTransformer [29] and TabNet [3], differentiable decision trees with NODE [50], and weight inference with Diet Networks [54]. We also attempted FT-Transformer [18], but it experienced out of memory issues on all datasets due to the large number of features. Finally, we consider graph regularization methods which also have access to the knowledge graph including GraphNet [20], NC LASSO [40], and Network LASSO [23] (Appendix E).

**Fair Comparison of PLATO with Baselines.** To ensure a fair comparison with baselines, we follow evaluation protocols in recent tabular benchmarks [19, 18]. We conduct a random search with 500 configurations of every model (including PLATO) on every dataset across a broad range of hyperparameters (Appendix A). We split data with a 60/20/20 training, validation, test split. All results are computed across 3 data splits and 3 runs of each model in each data split. We report the mean and standard deviation of the Pearson correlation (PearsonR) between $\mathbf{y}$ and $\hat{\mathbf{y}}$ across runs and splits on the test set. Each model is run on a GeForce RTX 2080 TI GPU.

Table 4: **PLATO's performance with an incomplete knowledge graph.**

| Fraction of edges in KG | PearsonR |
|---|---|
| 100% | $0.583 \pm 0.019$ |
| 90% | $0.570 \pm 0.017$ |
| 70% | $0.537 \pm 0.044$ |
| 50% | $0.412 \pm 0.011$ |

Table 5: **PLATO's MLP layers** $2, \ldots, L$ **with trainable weights are useful for performance.**

| Model | Description | PearsonR |
|---|---|---|
| PLATO | MLP with first layer weights inferred | $0.583 \pm 0.019$ |
| PLATO-LR | Linear regression with weights inferred | $0.550 \pm 0.020$ |

Table 6: **PLATO's performance is competitive with baselines when** $d \sim n$**.** For every dataset, the best overall model is in **bold** and the second best model is underlined.

| Dataset | | ME | BC | SCLC | NSCLC |
|---|---|---|---|---|---|
| # of features $d$ | | 19,902 | 18,261 | 18,437 | 18,308 |
| # of samples $n$ | | 10,064 | 10,101 | 10,712 | 16,730 |
| $d/n$ | | 2.0 | 1.8 | 1.7 | 1.1 |
| Classic Stat ML | Ridge | $0.566_{\pm 0.008}$ | $0.483_{\pm 0.008}$ | $0.604_{\pm 0.057}$ | $0.679_{\pm 0.008}$ |
| Dim. Reduct. | PCA | $0.239_{\pm 0.310}$ | $0.233_{\pm 0.294}$ | $0.284_{\pm 0.274}$ | $0.645_{\pm 0.000}$ |
| Feat. Select. | LASSO | $0.667_{\pm 0.000}$ | $0.633_{\pm 0.000}$ | $0.669_{\pm 0.000}$ | $0.637_{\pm 0.000}$ |
| | STG | $0.676_{\pm 0.000}$ | $0.643_{\pm 0.000}$ | $0.668_{\pm 0.000}$ | $0.646_{\pm 0.000}$ |
| Decision Tree | XGBoost | $\mathbf{0.875}_{\pm \mathbf{0.000}}$ | $\underline{0.826}_{\pm 0.000}$ | $\underline{0.878}_{\pm 0.000}$ | $\mathbf{0.843}_{\pm \mathbf{0.000}}$ |
| Graph Reg. | GraphNet | $0.675_{\pm 0.047}$ | $0.723_{\pm 0.026}$ | $0.742_{\pm 0.039}$ | $0.627_{\pm 0.042}$ |
| | NC LASSO | $0.733_{\pm 0.016}$ | $0.730_{\pm 0.027}$ | $0.793_{\pm 0.009}$ | $0.746_{\pm 0.023}$ |
| | Network LASSO | $0.401_{\pm 0.034}$ | $0.451_{\pm 0.022}$ | $0.417_{\pm 0.074}$ | $0.465_{\pm 0.034}$ |
| Param. Infer. | Diet | $0.105_{\pm 0.000}$ | $0.037_{\pm 0.000}$ | $-0.050_{\pm 0.000}$ | $0.002_{\pm 0.000}$ |
| Tabular DL | MLP | $0.487_{\pm 0.131}$ | $0.508_{\pm 0.061}$ | $0.537_{\pm 0.061}$ | $0.573_{\pm 0.005}$ |
| | NODE | $0.870_{\pm 0.000}$ | $0.420_{\pm 0.169}$ | $0.801_{\pm 0.102}$ | $0.487_{\pm 0.197}$ |
| | TabTransformer | $0.305_{\pm 0.028}$ | $0.010_{\pm 0.000}$ | $0.288_{\pm 0.203}$ | $0.503_{\pm 0.187}$ |
| | TabNet | $0.667_{\pm 0.002}$ | $0.624_{\pm 0.001}$ | $0.657_{\pm 0.004}$ | $0.647_{\pm 0.000}$ |
| Ours | PLATO | $\mathbf{0.875}_{\pm \mathbf{0.004}}$ | $\mathbf{0.844}_{\pm \mathbf{0.003}}$ | $\mathbf{0.883}_{\pm \mathbf{0.002}}$ | $\underline{0.839}_{\pm 0.000}$ |

## 4.1 Results

**PLATO outperforms statistical and deep baselines when** $d \gg n$**.** PLATO outperforms all baselines across all 6 datasets with $d \gg n$ (Table 1). PLATO achieves the largest improvement on the PDAC dataset, improving by 10.19% vs. XGBoost, the best baseline for PDAC (0.400 vs. 0.363). While PLATO achieves the strongest performance across all 6 datasets, the best performing baseline varies across datasets. Ridge Regression is the strongest baseline for BRCA, LASSO for CM and CRC, XGBoost for PDAC and CH, and TabTransformer for MNSCLC. The remaining baselines are not the strongest baseline for any dataset. We also find that the performance of a specific baseline depends largely on the dataset. TabTransformer, for example, is the best baseline for the MNSCLC dataset but the worst baseline for the CH dataset. The rank order of all models on all datasets is Appendix D.

**PLATO's performance depends on updating feature embeddings with a trainable message-passing function.** PLATO infers the weights $\hat{\Theta}^{[1]}$ in the first layer of a MLP $\mathcal{F}$ by using feature embeddings which contain prior information about the input features. PLATO first pretrains general feature embeddings $\mathbf{M} \in \mathbb{R}^{d \times c}$. PLATO then updates the feature embeddings to $\mathbf{Q} \in \mathbb{R}^{d \times c}$ with a trainable message-passing function. We test whether updating the feature embeddings based on the trainable message-passing function is necessary by evaluating PLATO's performance on the BRCA dataset in three configurations (Table 2). The default configuration uses the updated feature embeddings $\mathbf{Q}$ generated by the message-passing function to infer $\hat{\Theta}^{[1]}$ according to $\hat{\Theta}^{[1]}_j = \mathcal{B}(\mathbf{Q}_j | \mathbf{X}_i)$. The second configuration uses the general feature embeddings $\mathbf{M}$ instead of $\mathbf{Q}$ to infer $\hat{\Theta}^{[1]}$ according to $\hat{\Theta}^{[1]}_j = \mathcal{B}(\mathbf{M}_j)$. The third configuration does not use feature embeddings and thus ablates to a standard MLP. Using general feature embeddings $\mathbf{M}$ improves over not using feature embeddings at all (0.522 vs. 0.240). Using feature embeddings $\mathbf{Q}$ that are generated by the trainable message-passing function further improves performance (0.583 vs. 0.522). Thus, updating the feature embeddings to $\mathbf{Q}$ based on the trainable message-passing function is key to PLATO's performance.

**PLATO's performance depends on both feature nodes and broader knowledge nodes in the auxiliary KG.** PLATO relies on an auxiliary KG $G$ which contains information describing input features and the broader domain. Information describing input features is represented as feature nodes while information describing the broader domain is represented as other nodes in $G$ (Methods 3.1). To test the relative importance of the feature information in $G$ vs. the broader domain information, we measured the performance of PLATO on the BRCA dataset in two KG configurations: PLATO with the full KG (*i.e.* both the feature nodes and the broader domain nodes) and PLATO with a "feature-only KG" (*i.e.* an induced subgraph on only the feature nodes) (Table 3). We also compare to a "No KG" configuration in which PLATO does not have access to the KG. Without auxiliary information describing the input features or the broader domain, PLATO is ablated to a standard MLP.

We find that both the feature nodes and the broader knowledge nodes are important for PLATO's performance. Using the "feature-only KG" configuration of PLATO improves performance vs the "no KG" configuration (0.539 vs 0.240). Using the "full KG" configuration further improves performance vs the "feature-only KG" configuration (0.583 vs 0.539). PLATO's performance thus relies on both the feature information and the broader domain information in the KG.

**PLATO's performance with an incomplete knowledge graph.** All KGs are incomplete since there is undiscovered knowledge. PLATO thus uses low-dimensional embeddings from KG embedding approaches [72, 78, 65] which are designed to account for missing information, enabling predictive performance even with missing edges. We conduct an ablation study to assess PLATO's robustness to missing edges in the KG. We randomly remove edges from the KG and measure PLATO's performance on the BRCA dataset. We observe that with only 50% of the KG's edges, PLATO still has 71% of the performance as PLATO with 100% of the KG's edges (0.412 vs. 0.583) (Table 4).

**The importance of MLP layers** $2, \ldots, L$**, the layers with trainable weights, for PLATO.** PLATO is a MLP in which the weights in the first layer are inferred from the knowledge graph (KG) but the weights in the remaining layers $2, \ldots, L$ are trained normally. We conduct an ablation study to determine whether MLP layers $2, ..., L$ are necessary for PLATO's performance or whether the first layer of inferred weights are sufficient. Note that a single layer of inferred weights in PLATO is equivalent to a linear regression in which the weights are inferred from the KG. We thus compare PLATO to PLATO-LR, a linear regression in which the weights are inferred from the KG (Table 5). PLATO 's standard configuration outperforms PLATO-LR on the BRCA dataset (0.583 vs. 0.550). Therefore, layers $2, \ldots, L$ of the MLP are important for PLATO's performance.

**For datasets with** $d \sim n$**, PLATO is competitive with baselines.** Finally, we test PLATO's performance for datasets with $d \sim n$. We test 4 datasets with $d \sim n$ ranging from $\frac{d}{n} = 1.1$ to 2.0 (Table 6). We find that on 4 datasets with $d \sim n$, PLATO is competitive with the best baseline, XGBoost, but does not improve performance substantially. PLATO's stronger performance for datasets with $d \gg n$ than for datasets with $d \sim n$ is justified. PLATO's key idea is to include auxiliary information describing the input features. Auxiliary information is likely to help performance the most in settings with the least labeled data (*i.e.* $d \gg n$). When $d \sim n$, auxiliary information is less helpful since the tabular dataset may already have enough information to train a strong predictive model. We further find that XGBoost is the strongest baseline for all datasets with $d \sim n$, in contrast to XGBoost's varied performance on the datasets with $d \gg n$ (Table 1).

## 5   Discussion

PLATO achieves strong performance on tabular data when $d \gg n$ by using an auxiliary KG describing input features to regularize a multilayer perceptron (MLP) . Across 6 datasets, PLATO outperforms 13 state-of-the-art baselines by up to $10.19\%$. Ablations demonstrate the importance of PLATO's trainable message-passing function, of including nodes in the KG that don't represent input features but instead represent domain information, and of the layers in the MLP whose weights are trained directly rather than inferred. We also test PLATO's robustness to missing information in the KG. PLATO has several limitations. First, PLATO matches but does not improve the performance of baselines for high-dimensional datasets with more samples (*i.e.* $d \sim n$). Second, PLATO depends on the existence of an auxiliary KG of domain information. Overall, PLATO enables tabular deep learning when $d \gg n$ by using an auxiliary KG of domain information describing input features.

## Acknowledgements

We thank Maria Brbic, Michael Moor, Kaidi Cao, Weihua Hu, Rajas Bansal, Qian Huang, and Hamed Nilforoshan for discussions and for providing feedback on our manuscript. C.R. is supposed by a Siebel Fellowship, a National Science Foundation Graduate Research Fellowship under Grant No. DGE-1656518, and a Stanford Enhancing Diversity in Graduate Education (EDGE) Fellowship. H.R. is supposed by the Masason Foundation, an Apple Fellowship, and a Baidu Scholarship. We also gratefully acknowledge the support of DARPA under Nos. HR00112190039 (TAMI), N660011924033 (MCS); ARO under Nos. W911NF-16-1-0342 (MURI), W911NF-16-1-0171 (DURIP); NSF under Nos. OAC-1835598 (CINES), OAC-1934578 (HDR), CCF-1918940 (Expeditions), NIH under No. 3U54HG010426-04S1 (HuBMAP), Stanford Data Science Initiative, Wu Tsai Neurosciences Institute, Amazon, Docomo, GSK, Hitachi, Intel, JPMorgan Chase, Juniper Networks, KDDI, NEC, and Toshiba. The content is solely the responsibility of the authors and does not necessarily represent the official views of the funding entities.

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

# A Evaluation protocol and hyperparameter ranges

To ensure a fair comparison with baselines, we follow evaluation protocols outlined in tabular benchmarks [19, 18]. We conduct a random search with 500 configurations of every model (including PLATO) on every dataset across a broad range of hyperparameters. We base the hyperparameter ranges on the ranges used in prior tabular learning benchmarks [19, 18] and the ranges mentioned in the original papers of the methods. Hyperparameter ranges for PLATO are given in Table 7. Hyperparameter ranges for baseline methods are given in Table 8.

| Module in PLATO | Hyperparameter | Range |
|---|---|---|
| General | Learning rate
Batch size
L2 | LogUniform(1e-4, 5e-3)
[16, 32, 64]
0, LogUniform(1e-5, 1e-2) |
| KG $\mathcal{H}$ | Embedding dimension $c$
Embedding model | 200
ComplEx |
| Message Passing (MP) $\mathcal{Q}$ | # Rounds $R$
$\beta$
Hidden dimension in $\mathcal{A}$ | 2
LogUniform(1e-4, 1e-1)
UniformInt(16, 512) |
| Weight Inference $\mathcal{B}$ | # Layers
Hidden dimension | UniformInt(2, 6)
UniformInt(16, 512) |
| Layers $2, \ldots, L$ in MLP $\mathcal{F}$ | # Layers $L$
Hidden dimension | UniformInt(2, 6)
UniformInt(16, 512) |

Table 7: **Hyperparameter ranges used for PLATO.**

| Model | Hyperparameter | Range |
|---|---|---|
| LASSO | L1 | LogUniform(1E-4, 10) |
| Ridge | L2 | LogUniform(1E-4, 10) |
| XGBoost | n-estimators | UniformInt(1,2000) |
| | Max depth | UniformInt(3, 10) |
| | Min weight | LogUniform(1E-8,1E5) |
| | Subsample | Uniform(0.5, 1) |
| | Learning rate | LogUniform(1E-5,1) |
| | Col sample by level | Uniform(0.5, 1) |
| | Col sample by tree | Uniform(0.5, 1) |
| | Gamma | 0, LogUniform(1E-8, 1E2) |
| | Lambda | 0, LogUniform(1E-8, 1E2) |
| | Alpha | 0, LogUniform(1E-8, 1E2) |
| | Booster | "gbtree" |
| | Early-stopping-rounds | 50 |
| | Iterations | 100 |
| PCA | Number of PCA Components | UniformInt(2,1000) |
| STG | Hidden dimension | UniformInt(10, 500) |
| | Number of layers | UniformInt(1, 5) |
| | Activation | [Tanh, Relu, Sigmoid] |
| | Learning rate | LogUniform(1e-4, 1e-1) |
| | Sigma | Uniform(0.001, 2) |
| | Lambda | LogUniform(1e-3, 10) |
| MLP | Number of layers | UniformInt(1, 8) |
| | Hidden dimension | UniformInt(1, 512) |
| | Dropout | 0, Uniform([0,0.5]) |
| | Learning rate | LogUniform(1e-5, 1e-2) |
| | L2 | 0, LogUniform(1e-6, 1e-3) |
| TabNet | Decision Steps | UniformInt(3, 10) |
| | Layer size | 2, 4, 8, 16, 32, 64 |
| | Relaxation factor | Uniform[1, 2] |
| | Sparsity loss weight | LogUniform[1e-6, 1e-1] |
| | Decay rate | Uniform[0.4, 0.95] |
| | Decay steps | 100, 500, 2000 |
| | Learning rate | Uniform(1e-3, 1e-2) |
| | Iterations | 100 |
| TabTransformer | Embedding dimension | 4, 8, 16, 32, 64, 128 |
| | Number of heads | UniformInt(1, 10) |
| | Number of attention blocks | UniformInt(1, 12) |
| | Attention dropout rate | Uniform(0, 0.5) |
| | Add norm dropout | Uniform(0, 0.5) |
| | Transformation activation | [Tanh, Relu, LeakyReLU] |
| | L2 | LogUniform(1e-6, 1e-1) |
| | Learning rate | LogUniform(1e-6, 1e-3) |
| | FF dropout | Uniform(0, 0.5) |
| | FF hidden multiplier | 1, 2, 3, 4, 5, 6, 7, 8, 9, 10 |
| | Out FF activation | [Tanh, Relu, LeakyReLU] |
| | Out FF dropout | Uniform(0, 0.5) |
| NODE | Learning rate | LogUniform(1e-5, 1) |
| | Number of layers | UniformInt(1, 10) |
| | Number of trees | UniformInt(2, 2048) |
| | Depth | UniformInt(1, 10) |
| | Embedding choice | $X^T$, random |

Diet Network

|  | Number of layers | UniformInt(1, 8) |
|  | Hidden dimension | UniformInt(1, 512) |
|  | Dropout | 0, Uniform([0,0.5]) |
|  | Learning rate | LogUniform(1e-5, 1e-2) |
|  | L2 | 0, LogUniform(1e-6, 1e-3) |
| GraphNet | Hidden dimension | UniformInt(1, 512) |
|  | Learning rate | LogUniform(1e-5, 1e-2) |
|  | $\lambda$ | 0, LogUniform(1e-5, 1e2) |
|  | L1 coefficient | 0, LogUniform(1e-5, 1e2) |
| NC Lasso | Hidden dimension | UniformInt(1, 512) |
|  | Learning rate | LogUniform(1e-5, 1e-2) |
|  | $\lambda$ | 0, LogUniform(1e-5, 1e2) |
|  | L1 coefficient | 0, LogUniform(1e-5, 1e2) |
| Network Lasso | Hidden dimension | UniformInt(1, 512) |
|  | Learning rate | LogUniform(1e-5, 1e-2) |
|  | $\lambda$ | 0, LogUniform(1e-5, 1e2) |
|  | L1 coefficient | 0, LogUniform(1e-5, 1e2) |

Table 8: **Hyperparameter range for all baselines.**

## B   Graph classification approaches

Graph classification models are not appropriate for PLATO's setting. In graph classification models, every input sample is a graph with node attributes, and a model must make a prediction for that graph. The PLATO problem setting breaks key assumptions made by typical graph classification models. First, graph classification models assume that different samples correspond to different graphs [81, 28, 27]. However, in PLATO every sample corresponds to the exact same graph. There is a single background knowledge graph for all samples, and every sample has input features that correspond to the exact same nodes within the knowledge graph. Second, graph classification approaches typically assume that every node in an input graph has a node attribute [81, 28, 27]. However, in PLATO only a small subset of the nodes in the knowledge graph have measured feature values. Finally, graph classification approaches typically assume small graphs: the largest graph classification task in the Open Graph Benchmark has only 244 nodes [27]. However in PLATO, the knowledge graph contains 108,447 and the smallest dataset has 12,932 features corresponding to nodes.

## C   PLATO's performance across node embedding methods for pre-training the feature embeddings

We conduct an ablation study to assess how PLATO's performance depends on the node embedding method used to pre-train the feature embeddings (Methods 3.4). We test three shallow node embedding methods for knowledge graphs which are scalable and prominent: TransE [72], DistMult [78], and ComplEx [65]. We find that PLATO's performance is similar across TransE, DistMult, and ComplEx (Table 9). More generally, PLATO makes no assumption about what type of self-supervised node embedding method is used to pre-train the feature embeddings. The self-supervised embedding step is simply a module that pre-trains feature embeddings which are then passed to the message passing and weight inference modules of PLATO.

| KG Node Embedding Method | PearsonR (Test) on BRCA Dataset |
|---|---|
| TransE | $0.582 \pm 0.025$ |
| DistMult | $0.575 \pm 0.011$ |
| ComplEx | $0.583 \pm 0.019$ |

Table 9: **PLATO's performance is consistent across knowledge graph node embedding methods.**

## D   Rank ordering of methods for datasets with $d \gg n$

In Table 10, we show the rank order performance of all models on all $d \gg n$ datasets. We find that PLATO exhibits consistent and strong performance while the performance of the baselines depends on the specific $d \gg n$ dataset. For example, TabTransformer is the second best performing of all models on the MNSCLC dataset but the worst performing of all models on the PDAC and CH datasets. Similarly, XGBoost is the second best performing of all models on PDAC but only the tenth best performing of all models on BRCA. The baselines with the most stable performance are LASSO and Ridge Regression which rank consistently between the second and eighth best of all models.

Table 10: **For datasets with $d \gg n$, PLATO exhibits consistent and strong performance.** By contrast, the performance of the baselines varies with each dataset. For every dataset, the rank order of performance from Table 1 is shown. The best overall model is in **bold** and the second best model is underlined.

| Dataset | | MNSCLC | CM | PDAC | BRCA | CRC | CH |
|---|---|---|---|---|---|---|---|
| # of features $d$ / # of samples $n$ | | 52.2 | 46.1 | 40.3 | 28.2 | 22.6 | 19.7 |
| Classic Stat ML | Ridge | 8 | 3 | 4 | 2 | 3 | 5 |
| Dim. Reduct. | PCA | 7 | 12 | 10 | 7 | 10 | 8 |
| Feat. Select. | LASSO | 6 | 2 | 3 | 5 | 2 | 4 |
| | STG | 9 | 4 | 8 | 4 | 7 | 7 |
| Decision Tree | XGBoost | 14 | 9 | 2 | 10 | 5 | 2 |
| Graph Reg. | GraphNet | 5 | 7 | 9 | 8 | 12 | 3 |
| | NC LASSO | 4 | 5 | 5 | 6 | 9 | 6 |
| | Network LASSO | 3 | 8 | 12 | 9 | 11 | 9 |
| Param. Infer. | Diet | 13 | 13 | 6 | 12 | 13 | 11 |
| Tabular DL | MLP | 10 | 6 | 7 | 11 | 4 | 12 |
| | NODE | 12 | 10 | 11 | 3 | 6 | 10 |
| | TabTransformer | 2 | 11 | 14 | 13 | 8 | 14 |
| | TabNet | 11 | 14 | 13 | 14 | 14 | 13 |
| Ours | PLATO | **1** | **1** | **1** | **1** | **1** | **1** |

Table 11: **For datasets with $d \sim n$, PLATO is competitive with baselines.** For XGBoost is consistently the strongest baseline. For every dataset, the rank order of performance from Table 11 is shown. The best overall model is in **bold** and the second best model is underlined.

| Dataset | | ME | BC | SCLC | NSCLC |
|---|---|---|---|---|---|
| # of features $d$ / # of samples $n$ | | 2.0 | 1.8 | 1.7 | 1.1 |
| Classic Stat ML | Ridge | 9 | 9 | 9 | 4 |
| Dim. Reduct. | PCA | 13 | 12 | 13 | 7 |
| Feat. Select. | LASSO | 7.5 | 6 | 6 | 8 |
| | STG | 5 | 5 | 7 | 6 |
| Decision Tree | XGBoost | **1.5** | 2 | 2 | **1** |
| Graph Reg. | GraphNet | 6 | 4 | 5 | 9 |
| | NC LASSO | 4 | 3 | 4 | 3 |
| | Network LASSO | 11 | 10 | 11 | 13 |
| Param. Infer. | Diet | 14 | 13 | 14 | 14 |
| Tabular DL | MLP | 10 | 8 | 10 | 10 |
| | NODE | 3 | 11 | 3 | 12 |
| | TabTransformer | 12 | 14 | 12 | 11 |
| | TabNet | 7.5 | 7 | 8 | 5 |
| Ours | PLATO | **1.5** | **1** | **1** | 2 |

# E    Graph regularization baselines

We test the state-of-the-art graph regularization baselines GraphNet [20], Network-Constrained LASSO [40], and Network LASSO [23]. The graph regularization baselines can only consider a homogeneous graph with only features as nodes and a single edge type. For the graph regularization baselines, we thus induce a subgraph between feature nodes from the knowledge graph and collapse all edge types between feature nodes into a single edge type. In this context, GraphNet, Network-constrained LASSO, and Network LASSO correspond to a LASSO model with a mean-squared error loss and a graph regularization penalty. Let $\lambda$ be the graph regularization coefficient, $j$ and $k$ be two input features, let $E$ be the set of edges in the graph, let $\mathbf{\Theta} \in \mathbb{R}^d$ be the weights of the linear regression for $d$ input features, and let $D_j$ be the degree of feature node $j$. The graph regularization penalty for GraphNet is $\lambda \sum_{j,k \in E} (\mathbf{\Theta}_j - \mathbf{\Theta}_k)^2$, the penalty for Network-constrained LASSO is $\lambda \sum_{j,k \in E} (\frac{\mathbf{\Theta}_j}{\sqrt{D_j}} - \frac{\mathbf{\Theta}_k}{\sqrt{D_k}})^2$, and the penalty for Network LASSO is $\sum_{j,k \in E} |\mathbf{\Theta}_j - \mathbf{\Theta}_k|$.

# F    Number of trainable weights in PLATO vs. a multilayer perceptron

Table 12: **PLATO drastically reduces the number of trainable weights compared to a multilayer perceptron (MLP) across all of the datasets.** The number of trainable weights in the best model from the hyperparameter sweep is shown for each dataset.

| Model | MNSCLC | CM | PDAC | BRCA | CRC | CH | ME | BC | SCLC | NSCLC |
|---|---|---|---|---|---|---|---|---|---|---|
| MLP | 429665 | 416961 | 820097 | 425217 | 200529 | 589761 | 586945 | 296113 | 298929 | 594209 |
| PLATO | 17154 | 42498 | 32066 | 17154 | 28386 | 61890 | 17154 | 28386 | 32066 | 17154 |

# G    Dataset Details

We use 6 datasets with $d \gg n$ and 4 datasets with $d \sim n$ [16, 17, 30, 79]. In all datasets, a machine learning model must predict the response of a cell line or a mouse tumor model to a drug. As input, the model considers a tabular dataset. In the tabular dataset, every row corresponds to a specific cell line or mouse tumor model. Every column corresponds to a gene name. Every value corresponds to the amount of that gene in the cell line or in the mouse tumor (*i.e.* gene expression). In practice, the number of genes is large for all tasks and the number of cell lines or mouse tumor models is comparatively small (*i.e.* $d \gg n$). For every row, the model also takes as input a fixed feature vector corresponding to the drug (*i.e.* a 200-dimensional ComplEx [65] embedding of the drug node in the knowledge graph). The output label is the response of the cell line (*i.e.* ln-ic50) [30, 79, 17] or mouse tumor model (*i.e.* minimum average percent tumor growth "min-avg-pct-tumor-growth") to the drug [16]. Data is available at `https://github.com/snap-stanford/plato`.

Gene expression datasets were pre-processed following a standard process in [48]. Briefly, gene expression values underwent TMM normalization and log transformation (*i.e.* $\log(x + 1)$). Values were made to have zero mean and unit standard deviation. Dataset abbreviations are breast carcinoma (BC) [30, 79, 17], breast carcinoma (BRCA) [16], chondrosarcoma (CH) [30, 79, 17], colorectal cancer (CRC) [16], cutaneous melanoma (CM) [16], melanoma (ME) [30, 79, 17], non-small cell lung carcinoma (MNSCLC) [16], non-small cell lung carcinoma (NSCLC) [30, 79, 17], pancreatic ductal carcinoma (PDAC) [16], and small cell lung carcinoma (SCLC) [30, 79, 17].

# H  Knowledge Graph Details

We compile a general biomedical knowledge graph from prior studies [44, 35, 38, 56, 63, 74, 75] to use across all datasets. A schematic of the KG is in Supplementary Figure 2. A detailed breakdown of relation types is in Supplementary Table 13. The knowledge graph is available at `https://github.com/snap-stanford/plato`.

The knowledge graph contains 108,447 total nodes, including 7,975 drugs, 18,370 diseases, 11,447 phenotypes, 22,319 genes, 11,153 molecular functions, 28,748 biological processes, and 4,184 cellular components. Every gene and every drug in every dataset is present as a node in the knowledge graph. The knowledge graph also contains 3,066,156 edges with 99 distinct relation types. The remaining node types and their relationships serve as broader domain knowledge.

Edges between drug nodes and gene/protein nodes were derived from Drugbank [75], Gao [16], and the Genomics of Drug Sensitivity in Cancer [79, 30, 17]. Edges between diseases and genes/proteins were derived from DisGeNet [5]. Edges between diseases and phenotypes were derived from the Human Phenotype Ontology [35]. Edges between drugs and diseases were derived from the Multiscale Interactome [56]. Edges between drugs and side effects were derived from SIDER [38]. Edges between genes/proteins and other genes/proteins were derived from BioGRID [49], [55], the Database of Interacting Proteins [57], [44], [47], [53], [82], [69], and STRING [63]. Finally, edges from genes/proteins to molecular functions, biological processes, and cellular components as well as edges between molecular functions, biological processes, and cellular components were derived from the Gene Ontology [9].

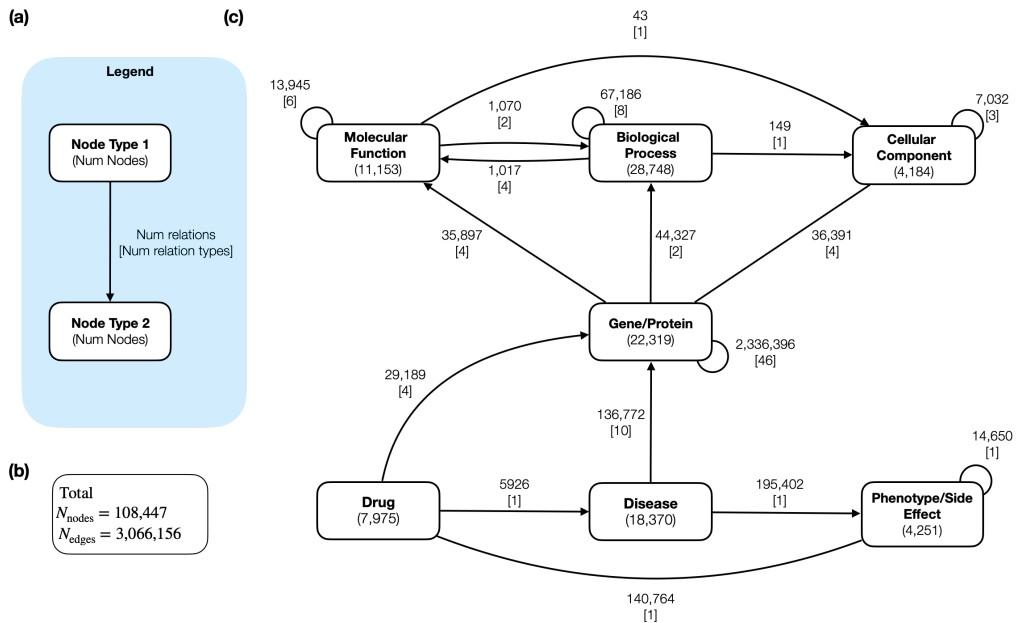

Figure 2: **Knowledge graph as a unified knowledge backbone.** We constructed a knowledge graph as a unified knowledge backbone across all 6 datasets. (a) Legend. For each node type, the number of nodes is given in parentheses. Between node types, the number of edges and the number of relation types are given. (b) Number of total nodes and edges across entire knowledge graph. (c) Visual schematic of knowledge graph across each node type.

| Head type | Relation | Tail type | # edges |
|---|---|---|---|
| BiologicalProcess | EndsDuring | BiologicalProcess | 1 |
| BiologicalProcess | HappensDuring | BiologicalProcess | 8 |
| BiologicalProcess | HasPart | BiologicalProcess | 229 |
| BiologicalProcess | IsA | BiologicalProcess | 53015 |
| BiologicalProcess | NegativelyRegulates | BiologicalProcess | 2768 |
| BiologicalProcess | PartOf | BiologicalProcess | 5193 |
| BiologicalProcess | PositivelyRegulates | BiologicalProcess | 2756 |
| BiologicalProcess | Regulates | BiologicalProcess | 3216 |
| BiologicalProcess | OccursIn | CellularComponent | 149 |
| BiologicalProcess | HasPart | MolecularFunction | 173 |
| BiologicalProcess | NegativelyRegulates | MolecularFunction | 269 |
| BiologicalProcess | PositivelyRegulates | MolecularFunction | 274 |
| BiologicalProcess | Regulates | MolecularFunction | 301 |
| CellularComponent | HasPart | CellularComponent | 179 |
| CellularComponent | IsA | CellularComponent | 4863 |
| CellularComponent | PartOf | CellularComponent | 1990 |
| Disease | AlteredExpression | Gene | 7157 |
| Disease | Biomarker | Gene | 107160 |
| Disease | ChromosomalRearrangement | Gene | 162 |
| Disease | FusionGene | Gene | 166 |
| Disease | GeneticVariation | Gene | 15076 |
| Disease | GermlineCausalMutation | Gene | 4677 |
| Disease | ModifyingMutation | Gene | 10 |
| Disease | SomaticCausalMutation | Gene | 130 |
| Disease | SusceptibilityMutation | Gene | 441 |
| Disease | Therapeutic | Gene | 1793 |
| Disease | Has | Phenotype | 195402 |
| Drug | Treats | Disease | 5926 |
| Drug | Carries | Gene | 866 |
| Drug | Enzymes | Gene | 5382 |
| Drug | Targets | Gene | 19817 |
| Drug | Transports | Gene | 3124 |
| Drug | Has | Phenotype | 140764 |
| Gene | Associates | BiologicalProcess | 43857 |
| Gene | NotAssociates | BiologicalProcess | 470 |
| Gene | Associates | CellularComponent | 35306 |
| Gene | Colocalizes | CellularComponent | 914 |
| Gene | NotAssociates | CellularComponent | 160 |
| Gene | NotColocalizes | CellularComponent | 11 |
| Gene | Acetylation | Gene | 9 |
| Gene | Activation | Gene | 58502 |
| Gene | AdpRibosylation | Gene | 2 |
| Gene | Ampylation | Gene | 5 |
| Gene | Association | Gene | 18 |
| Gene | Binary | Gene | 56565 |
| Gene | Binding | Gene | 287641 |
| Gene | Catalysis | Gene | 344801 |
| Gene | Cleavage | Gene | 22 |
| Gene | Complexes | Gene | 62552 |
| Gene | CovalentBinding | Gene | 52 |
| Gene | Deacetylation | Gene | 8 |
| Gene | Demethylation | Gene | 6 |
| Gene | Dephosphorylation | Gene | 26 |
| Gene | Deubiquitination | Gene | 18 |

| | | | |
|---|---|---|---|
| Gene | DirectInteraction | Gene | 2904 |
| Gene | DisulfideBond | Gene | 5 |
| Gene | DosageGrowthDefect | Gene | 9 |
| Gene | DosageLethality | Gene | 112 |
| Gene | DosageRescue | Gene | 63 |
| Gene | Enzymatic | Gene | 2 |
| Gene | Expression | Gene | 188 |
| Gene | GeneticInterference | Gene | 32 |
| Gene | Hydroxylation | Gene | 26 |
| Gene | Inhibition | Gene | 20108 |
| Gene | Kinase | Gene | 11960 |
| Gene | Literature | Gene | 174162 |
| Gene | Metabolic | Gene | 10646 |
| Gene | Methylation | Gene | 25 |
| Gene | NegativeGenetic | Gene | 3449 |
| Gene | OxidoreductaseActivityElectronTransferAssay | Gene | 2 |
| Gene | PhenotypicEnhancement | Gene | 209 |
| Gene | PhenotypicSuppression | Gene | 214 |
| Gene | Phosphorylation | Gene | 166 |
| Gene | Phosphotransfer | Gene | 1 |
| Gene | PhysicalAssociation | Gene | 824164 |
| Gene | PositiveGenetic | Gene | 2331 |
| Gene | PostTranslationalModification | Gene | 5306 |
| Gene | ProteinCleavage | Gene | 48 |
| Gene | PutativeSelfInteraction | Gene | 3 |
| Gene | Reaction | Gene | 400658 |
| Gene | Regulation | Gene | 2650 |
| Gene | Signaling | Gene | 65412 |
| Gene | SyntheticGrowthDefect | Gene | 407 |
| Gene | SyntheticLethality | Gene | 816 |
| Gene | SyntheticRescue | Gene | 91 |
| Gene | Associates | MolecularFunction | 35012 |
| Gene | Contributes | MolecularFunction | 596 |
| Gene | NotAssociates | MolecularFunction | 285 |
| Gene | NotContributes | MolecularFunction | 4 |
| MolecularFunction | PartOf | BiologicalProcess | 1068 |
| MolecularFunction | Regulates | BiologicalProcess | 2 |
| MolecularFunction | OccursIn | CellularComponent | 43 |
| MolecularFunction | HasPart | MolecularFunction | 204 |
| MolecularFunction | IsA | MolecularFunction | 13631 |
| MolecularFunction | NegativelyRegulates | MolecularFunction | 42 |
| MolecularFunction | PartOf | MolecularFunction | 11 |
| MolecularFunction | PositivelyRegulates | MolecularFunction | 27 |
| MolecularFunction | Regulates | MolecularFunction | 30 |
| Phenotype | IsA | Phenotype | 14650 |

Table 13: **Knowledge graph relations between node types.**

