# OpenReview forum: "High dimensional, tabular deep learning with an auxiliary knowledge graph"
_NeurIPS.cc/2023/Conference — NeurIPS 2023 poster_

### Official Review · Reviewer_dZQm · 2023-06-26

**Soundness:** 4 excellent
**Presentation:** 4 excellent
**Contribution:** 3 good
**Rating:** 7
**Confidence:** 4

**Summary:**

The authors present a method to leverage prior knowledge in the form of a knowledge graph for tabular deep learning, illustrating performance on a number of biological prediction problems in which features correspond to gene-level measurements. In this case, the knowledge graph is a biomedical knowledge graph which contains a node for each gene as well as nodes representing related biomedical concepts. A set of initial node embeddings are generated using a standard KG embedding approach. Once these are obtained, a set of node vectors for each sample is generated via message passing on the knowledge graph, making use of the sample values for each feature in the message passing step. These weights are then used as the first layer in a MLP.

The authors show that their method consistent achieves the best performance across a set of 6 data-sets where the number of features is significantly larger than the number of samples and is consistent with XGBoost when the number of features is comparable to the number of dimensions, suggesting the method could be a good default choice in cases where a relevant knowledge graph is available

The authors conduct a comprehensive ablation study to demonstrate the necessity of each component of their model in achieving the best prediction performance.

**Strengths:**

The approach is non-obvious and effective, while also being intuitive and clearly explained.The authors have combined ideas from knowledge graph embedding and graph-based regularisation to provide a state-of-the-art model for tabular problems in which a relevant knowledge graph is available. Given the abundance of prediction problems in the biomedical space and the accessibility of high quality knowledge graphs, I expect this method will see many applications there.

**Weaknesses:**

The authors claim PLATO is robust to missing edges in the knowledge graph. However, when 50% of edges have been removed the model is outperformed by multiple approaches on BRCA, the data-set on which this comparison was carried out. To make this claim, I think the authors need to show that the SOTA performance is maintained across all data-sets in the presence of missing edges.

**Questions:**

Can you include data for missing edges in the other 6 data-sets? Looking at Table 4, I am left wondering if the model will indeed outpeform all baselines when using 70% or 75% of the data

An interesting extension, or an opportunity for future work, could evaluate whether adding high-confidence predictions from ComplEx was sufficient to recover the performance drop at 50%.

**Limitations:**

The authors have addressed the key technical limitation: the availability of a good quality and comprehensive knowledge graph. The data presented in Table 4 could be reframed as an additional statement of this limitation - in the presence of an incomplete knowledge graph the performance is worse than simple baselines like Ridge.

I do not believe the model has potential negative societal impact

---

> ### Author Rebuttal · Authors · 2023-08-10
>
>
> > **Overall response**
>
> We **thank the reviewer** for their excellent review! The reviewer states that PLATO is **“non-obvious and effective”** while also being **“intuitive and clearly explained.”** The reviewer highlights PLATO’s novelty in “[combining] ideas from knowledge graph embedding and graph-based regularization to provide a **state-of-the-art model for tabular problems in which a relevant knowledge graph is available**.” Finally, the reviewer emphasizes that PLATO will **"see many applications" in the "biomedical space"** given “the **abundance of prediction problems**" and the "**accessibility of high quality knowledge graphs**.” We thank the reviewer for their comments which have strengthened the manuscript.
>
> ---
>
> > **The authors claim that PLATO is robust to missing edges in the knowledge graph. However, when 50% of the edges have been removed, the model is outperformed by multiple approaches on the BRCA dataset. To make this claim, I think the authors need to show that the SOTA performance is maintained across all data-sets in the presence of missing edges.**
>
> **We thank the reviewer for their comment and will revise the phrase “PLATO is robust to missing edges in the knowledge graph.”** We do not claim that PLATO is state-of-the-art on any of the datasets (BRCA or others) with only 50% or 75% of the edges in the knowledge graph. We only claim that PLATO outperforms the baselines on the 6 $ d \gg n$ datasets when using 100% of the edges which is shown in Table 1.
>
> The ablation analysis shown in Table 4 is meant to demonstrate how PLATO’s performance is affected by the removal of edges in the knowledge graph for the BRCA dataset. We agree that our initial phrase of “PLATO is robust to missing edges in the knowledge graph” should be clarified. We will revise the phrase to read “with 50% of the edges, PLATO retains 71% of the performance as PLATO with 100% of the edges on the BRCA dataset (Table 4).” We thank the reviewer for their comment which strengthens the manuscript!
>
> ---
>
> > **The authors test PLATO’s performance when 50% of the edges in the knowledge graph are removed. An interesting direction for future work would be to use a link prediction method to predict some of the missing edges and subsequently apply PLATO to improve performance.**
>
> We thank the reviewer for their question! When 50% of the edges in the knowledge graph are removed, it may be possible to first use a link prediction methodology to recover some of the missing edges and subsequently apply PLATO. We view this as an **exciting direction for future work** and will add it to the discussion section of our manuscript!
>
> ---
>
> > **The authors have addressed the key technical limitation: the availability of a good quality and comprehensive knowledge graph. The data presented in Table 4 could be reframed as an additional statement of this limitation.**
>
> We thank the reviewer for pointing out that we have already addressed the key technical limitation! Per the reviewer’s suggestion, **we will re-state the importance of knowledge graph in the ablation study in Table 4.**
>
> ---

---

> > ### Author Response · Authors · 2023-08-19
> > **Follow-up**
> >
> > We thank the reviewer for their time and thoughtful questions! We wanted to gently follow up before the discussion period closes. If our response has satisfied the reviewer, we would appreciate the reviewer considering raising their score. Thank you!

---

> > > ### Comment · Reviewer_dZQm · 2023-08-20
> > > **Response to author rebuttal**
> > >
> > > I thank the authors for their response. I don't choose to raise my score, but feel like the authors have responded to other reviews sufficiently to support acceptance

---

### Official Review · Reviewer_p8DV · 2023-07-03

**Soundness:** 2 fair
**Presentation:** 2 fair
**Contribution:** 3 good
**Rating:** 5
**Confidence:** 3

**Summary:**

The paper introduces PLATO, a novel machine learning approach for high-dimensional tabular data, where the number of features far exceeds the number of samples (d>>n). This scenario is often encountered in fields like physical sciences and biology. PLATO uses a knowledge graph (KG) to train the first layer of a multilayer perceptron (MLP), creating feature embeddings based on the information in the KG and a message-passing algorithm. This approach leverages auxiliary domain information to regularize the MLP, reducing the risk of overfitting. Benchmarking PLATO on six biomedical datasets revealed it outperformed 13 state-of-the-art baselines by up to 10.19%, demonstrating its effectiveness for the d>>n scenario.

**Strengths:**

The mechanism to induce KG regularization is novel. Studying KG integration into predictions seems understudied, while it has been done in specific contexts (e.g. gene interaction networks https://www.nature.com/articles/nmeth.4627, https://www.nature.com/articles/s42003-021-02622-z). The studies setting matters especially in a scientific context. A general method for knowledge graph integration could be very valuable to numerous scientific fields. The work is clearly written, with minor writing issues (especially in conclusion). It seems to provide most experimental details necessary.

An ablation of varying the fraction of the KG graph used clearly shows that the method is able to learn from increasing domain knowledge.


**Weaknesses:**

Baselines may not be as well tuned, e.g. use combinations of PCA + XGB. Hyperparameters might not be appropriate for the high dimensional setting, e.g. for trees.
A cross-validation procedure, e.g. 5-fold CV to find optimal hyperparameters on train and valid would be more optimal. Baselines with many hyperparameters to tune would profit more from this stronger evaluation. This is also what data-scientists would perform on such data, especially given the low N.
Its not an apples to apples comparison for all methods since domain knowledge is used for this approach. A simple baseline using domain knowledge + Baselines would be interesting, e.g. keeping only relevant genes in BRCA dataset based on prior knowledge. I believe significance values for genetic locations from prior studies are much more commonly known than graph connectivity. I understand this may still be hard to evaluate.


**Questions:**

Missing / Hard to understand where knowledge graphs come from. Were they created using the full data? Were they created using completely separate experiments? What was the method to create each? the supplement lists only the studies with no further details on how the graphs were obtained.

**Limitations:**

They adress some limitations. I would add that while PLATO uses domain knowledge, it only models one aspect of the domain knowledge provided by the graph: the similarity of nodes in the graph (Hope I correctly understand that?).

---

> ### Author Rebuttal · Authors · 2023-08-10
>
>
> > **Overall response**
>
> We thank the reviewer for pointing out that **“the mechanism to induce KG regularization is novel.”** The reviewer further notes that our study’s **“setting matters especially in a scientific context”** and that our **“general method for knowledge graph integration could be very valuable to numerous scientific fields.”** We have made multiple clarifications and strengthened the manuscript based on the reviewer’s comments. Thank you!
>
> ---
>
> > **How did the authors select the hyperparameter ranges for the baseline methods and the overall protocol to tune hyperparameters and evaluate PLATO compared to the baselines?**
>
> **We selected hyperparameter ranges for the baseline methods and the overall protocol to tune hyperparameters from recent tabular deep learning benchmarks [1, 2].** For each model, we select the hyperparameter range based on the range used in prior tabular learning benchmarks [1, 2] or the original papers. We then hyperparameter tune every model by conducting a random search with 500 configurations of every model on every dataset. We use the **same evaluation and tuning protocol on PLATO and on baselines**, thus ensuring a **fair comparison.**
>
> Ultimately, PLATO outperforms all baselines on 6 $ d \gg n$ datasets by up to 10.19% (Table 1). The evaluation protocol and hyperparameter ranges are further detailed in Appendix A, Table 7, and Table 8.
>
> [1] Grinsztajn et al. "Why do tree-based models still outperform deep learning on typical tabular data?." NeurIPS (2022).
>
>
> [2] Gorishniy et al. Revisiting deep learning models for tabular data. NeurIPS (2021).
>
> ---
>
> > **PLATO uses domain knowledge in addition to the tabular baselines to achieve its strong performance. Can the authors compare PLATO to a baseline method that also uses domain knowledge?**
>
> **We compare PLATO to three baseline methods that use domain knowledge.** GraphNet [1], Network-Constrained LASSO (NC LASSO) [2], and Network LASSO [3] are three graph regularization approaches that use the auxiliary knowledge graph in addition to the tabular dataset. In these graph regularization approaches, the feature relationships in the KG are used to smooth the weights in a regression model, thus using domain knowledge as suggested by the reviewer. **PLATO outperforms all graph regularization baselines by >19% (Rebuttal Table 1, Table 1).** We believe these graph regularization baselines are the most relevant comparisons as they use the same systematic prior knowledge as PLATO (e.g. the knowledge graph) rather than a separate prior knowledge source (e.g. hand-engineered features for a particular dataset).
>
> ---
>
> > **Where does the knowledge graph come from? Was it constructed from the tabular data or orthogonal information?**
>
> **PLATO’s knowledge graph comes from 6 prior databases which use information that is orthogonal to the tabular data.** To repeat, PLATO’s tabular datasets were not used to construct the auxiliary knowledge graph. Each of the 6 databases that we use to construct the overall knowledge compiles its edges from either relationships previously described in the scientific literature via literature curation (e.g. disease-symptom relationships) or through orthogonal scientific experiments which directly measure edges in the graph (i.e. protein-protein interactions via high-throughput experiments). We detail every database below, the source of it’s relationships, and the reference (numbers from article).
>
> | Database                               | Source                            | Reference in manuscript |
> |----------------------------------------|-----------------------------------|-----------|
> | Protein-protein interactions (Luck et al.)  | Orthogonal Scientific Experiments | [39]      |
> | Human Phenotype Ontology               | Literature Curation               | [31]       |
> | SIDER                                  | Literature Curation               | [34]       |
> | Multiscale Interactome                 | Literature Curation               | [46]       |
> | STRING                 | Literature Curation, Orthogonal Scientific Experiments   | [52]       |
> | DrugBank                               | Literature Curation               | [61, 62]       |
>
> ---
>
> > **As a limitation, the authors may consider including that PLATO only models one aspect of the domain knowledge provided by the graph: the similarity between nodes.**
>
> We thank the reviewer for their suggestion and will include this as a limitation in the revised version of the manuscript!
>
> ---

---

> > ### Comment · Reviewer_p8DV · 2023-08-10
> >
> > Thank you for addressing all my concerns, I find this a quite convincing rebuttal. I have also read the general reply and PDF.
> >
> > I did not check the evaluated baseline in detail and am relying on this being well implemented. This would be my only concern:
> > - XGB is isually very strong for Tab Data. In this work, it is doing very bad ( e.g. second to last on MNSCLC) while doing very well for other datasets. I find that a bit strange
> > - LASSO seems very strong beating e.g. GraphNet baselines that has domain knowledge - but maybe LASSO is just very strong?!

---

> > > ### Author Response · Authors · 2023-08-21
> > > **Response to p8DV Comment**
> > >
> > > > **Overall Response**
> > >
> > > We thank the reviewer for saying that we have provided **“a quite convincing rebuttal.”** We are grateful for the reviewer’s comments which we incorporate to strengthen the manuscript! If our response has satisfied the reviewer, we would appreciate the reviewer considering raising their score. Thank you!
> > >
> > > ---
> > >
> > > > **XGBoost is usually very strong for tabular datasets. Why does XGBoost perform well on most tabular datasets in this study but poorly on some?**
> > >
> > > **The reviewer is correct that XGBoost is very strong for tabular datasets, particularly those with more samples than features (e.g. $n > d$ or $n \gg d$).** Two recent tabular deep learning benchmarks demonstrate XGBoost’s strength when $n \gg d$ [1, 2]. Our manuscript further confirms XGBoosts’s strength on tabular data with a sufficient number of samples. For four datasets with $n \sim d$, XGBoost is consistently the strongest baseline and matches PLATO’s performance (Table 6, Supplementary Table 11), making our work consistent with prior literature.
> > >
> > > **However, PLATO is designed for tabular datasets with far more features than samples (e.g. $d \gg n$).** Prior work describing the strength of XGBoost for tabular datasets focuses on $n \gg d$ datasets and does not include $d \gg n$ datasets [1, 2, 3]. We find that when $d \gg n$, XGBoost’s performance, like that of all baselines, varies (Table 1, Supplementary Table 10). Nonetheless, PLATO demonstrates consistently strong performance when $d \gg n$, outperforming all baselines by up to 10.19%, highlighting PLATO’s contribution.
> > >
> > > ---
> > >
> > > > **LASSO appears to be strong. On a few datasets, it even beats the GraphNet baseline that uses domain knowledge. Is LASSO just that strong?**
> > >
> > > **As a feature selection method, LASSO is indeed known to be a strong baseline for datasets in which the number of features exceeds the number of samples (e.g. $d \gg n$) [4, 5].** Our manuscript confirms LASSO's strength: for four of the six $d \gg n$ datasets, LASSO outperforms GraphNet and the other graph regularization baselines, even though the graph regularization baselines use auxiliary domain knowledge while LASSO does not (Table 1).
> > >
> > > **However, LASSO outperforming the graph regularization baselines on four of the six $d \gg n$ datasets merely strengthens PLATO’s contribution.** PLATO uses domain knowledge but outperforms all baselines on all six $d \gg n$ datasets. Just using domain knowledge, as the graph regularization baselines do, is thus insufficient. The specific mechanism through which PLATO uses domain knowledge is critical (Methods). We thank the reviewer for their excellent observation and will add this discussion to the Appendix!
> > >
> > > ---
> > >
> > > > **References**
> > >
> > > [1] Gorishniy et al. “Revisiting deep learning models for tabular data.” NeurIPS (2021).
> > >
> > > [2] Grinsztajn et al. “Why do tree-based models still outperform deep learning on typical tabular data?” NeurIPS (2022).
> > >
> > > [3] Chen et al. "XGBoost: A scalable tree boosting system." Proceedings of the 22nd ACM SIGKDD International Conference on Knowledge Discovery and Data Mining (2016).
> > >
> > > [4] Tibshirani et al. "Regression shrinkage and selection via the lasso." Journal of the Royal Statistical Society Series B: Statistical Methodology (1996).
> > >
> > > [5] Liu et al. "Deep Neural Networks for High Dimension, Low Sample Size Data." IJCAI. 2017.

---

### Official Review · Reviewer_87H1 · 2023-07-07

**Soundness:** 2 fair
**Presentation:** 4 excellent
**Contribution:** 2 fair
**Rating:** 5
**Confidence:** 5

**Summary:**

The paper explores tabular ML tasks in scenarios where data has more features than samples and auxiliary structured data is available in the form of knowledge graphs. One of the central assumptions is that each feature corresponds to a node in the knowledge graph. A model called PLATO is proposed which combines a tabular MLP with the auxiliary knowledge graph, with the main idea that for features which correspond to similar nodes in the graph weight vectors in the first MLP layer should also be similar. The paper showcases performance gains of PLATO on 6 datasets compared to tabular models trained solely on the tabular data without leveraging the knowledge graph.

**Strengths:**

I would like to thank the authors for this paper. While I believe that it may not be quite ready for publication, I enjoyed reading the paper. The presentation is excellent and the ideas are explained well, it was definitely a fun read. Overall, I believe the paper has potential and I hope my feedback would help make the paper stronger.

Strengths:
- This paper explores an interesting and useful idea of leveraging auxiliary knowledge graphs to aid in tabular data tasks
- The presentation is excellent, the ideas are explained well, with great figures
- It is clear that the authors made a considerable effort to evaluate multiple tabular baselines
- The work is reproducible with the code included in the submission

**Weaknesses:**

I believe that a number of weaknesses need to be addressed before the paper can be published.

**Major:**

Graph perspective
- My main criticism is that it seems that PLATO is more of a method to learn graph embeddings which leverage auxiliary tabular data to solve subgraph classification or regression tasks rather than a tabular architecture regularized by a graph. This is a key distinction because it defines baselines that the model should be compared to. *That is, PLATO is more of a graph model and should be compared to graph models.*

 *Explanation for why PLATO is more a graph model than a tabular model*: As PLATO infers weight vectors in the first MLP layer which correspond to the input features from graph embeddings, it essentially means that that first MLP layer becomes simply a linear combination of graph-based feature embeddings with coefficients coming from feature value for a particular sample. This linear combination is then passed through a nonlinearity and into a downstream model (which in this case is all the other layers of MLP but it does not have to be just that). As the central assumption is that each feature has a corresponding graph node, this means that each sample can be represented as a subgraph with different weights on the subgraph nodes which correspond to feature value. So this becomes a subgraph classification/regression problem with the addition of those weights.

- As such, I encourage the authors to compare against suitable graph neural network baselines for the problem of subgraph classification/regression. One possible baseline is GCN-based node embeddings combined into subgraph embeddings as weighted combinations with weights coming from feature values.

Tabular perspective
- If we think of PLATO as a method for regularizing tabular architectures using a graph, then I see two major limitations: (1) PLATO is limited to MLP architectures only, (2) the central assumption is that *each* feature has a corresponding node in the graph is extremely limiting. If any of these assumptions can be relaxed, then it would be great to see corresponding experiments.
- It seems likely that the performance gains observed in Table 1 come solely from leveraging the knowledge graph and not as much from the strength of PLATO as a tabular architecture. Other models in Table 1 do not leverage the knowledge graph in any way. Table 3 highlights that performance gains likely primarily come from the knowledge graph as Full KG gives significantly better performance than feature-only KG. That is, the performance gains do not come from regularizing and avoiding overfitting in short and fat datasets, but from the additional information source.
- Ideally, PLATO should be compared against graph baselines, tabular baselines and some simple graph+tabular baselines to showcase the strength and justify the complexity of the PLATO architecture and design choices.

**Less major:**
- The paper claims to follow evaluation protocols from [1]. However, [1] used Bayesian HPO framework Optuna which is a stronger approach than random hyperparameter search.
- As mentioned in the paper, it is true that default FT-Transformer fails with out of memory errors on datasets with many features. However, the RTDL package implements Linformer strategy for efficient attention to address cases with many features. To be able to use it as a baseline model, I encourage the authors to explore the kv_compression_ratio hyperparameter from the FT-Transformer documentation https://yura52.github.io/rtdl/stable/api/rtdl.FTTransformer.html?highlight=linformer

**Related work issues:**
- A significant portion of related work on tabular deep learning is missing. For example, tabular transofmer architectures [2,3,4] could be added to the section Tabular deep learning methods. Additionally, while the authors say that "tabular deep learning methods have been developed for settings with far more samples than features" and "in the low-data regime with far more features than samples, the dominant approaches are still statistical", there are in fact tabular deep learning works which specifically address low data regimes through transfer learning [5]. There are also other deep tabular transfer learning architectures [6,7].


References:

[1] Gorishniy, Y., Rubachev, I., Khrulkov, V. and Babenko, A., 2021. Revisiting deep learning models for tabular data. Advances in Neural Information Processing Systems, 34, pp.18932-18943.

[2] Hollmann, N., Müller, S., Eggensperger, K. and Hutter, F., 2022. Tabpfn: A transformer that solves small tabular classification problems in a second. arXiv preprint arXiv:2207.01848.

[3] Somepalli, G., Goldblum, M., Schwarzschild, A., Bruss, C.B. and Goldstein, T., 2021. Saint: Improved neural networks for tabular data via row attention and contrastive pre-training. arXiv preprint arXiv:2106.01342.

[4] Kossen, J., Band, N., Lyle, C., Gomez, A.N., Rainforth, T. and Gal, Y., 2021. Self-attention between datapoints: Going beyond individual input-output pairs in deep learning. Advances in Neural Information Processing Systems, 34, pp.28742-28756.

[5] Levin, R., Cherepanova, V., Schwarzschild, A., Bansal, A., Bruss, C.B., Goldstein, T., Wilson, A.G. and Goldblum, M., 2022. Transfer learning with deep tabular models. arXiv preprint arXiv:2206.15306.

[6] Zhu, B., Shi, X., Erickson, N., Li, M., Karypis, G. and Shoaran, M., 2023. XTab: Cross-table Pretraining for Tabular Transformers. arXiv preprint arXiv:2305.06090.

[7] Wang, Z. and Sun, J., 2022. Transtab: Learning transferable tabular transformers across tables. Advances in Neural Information Processing Systems, 35, pp.2902-2915.

**Questions:**

1. I encourage the authors to compare against suitable graph neural network baselines for the problem of subgraph classification/regression. One possible baseline is GCN-based node embeddings combined into subgraph embeddings as weighted combinations with weights coming from feature values.
2. How can categorical features be handled by PLATO?
3. If PLATO can be used in cases where only some of the features correspond to nodes in the knowledge graph and other features don't, it would be great to see corresponding experiments.
4. It is mentioned on page 3, line 101 that PLATO "represents a new regularization mechanism". If PLATO can be applied as a regularization technique to architectures other than MLP, it would be great to see corresponding experiments.
5. Expanding on 1, as it is clear from Table 3 that the knowledge graph is a significant driver of performance gains as an additional information source, is it possible to compare against other tabular baselines which also leverage the KG information?

**Limitations:**

The authors did not address limitations or societal impact of the work.

---

> ### Author Rebuttal · Authors · 2023-08-10
>
> > **Overall response**
>
> We thank the reviewer for their **incredibly thoughtful review**! The reviewer says that the “**presentation is excellent**,” and the idea of “leveraging auxiliary KGs to aid in tabular data tasks” is “**interesting and useful.**”
>
> ---
>
> > **The first MLP layer is a linear combination of graph-based feature embeddings with coefficients coming from feature values for a particular sample. The PLATO problem setting should thus be seen as a subgraph classification/regression problem where every input sample corresponds to a subgraph and every node has a one-dimensional node attribute which corresponds to the feature value for that node. A GCN graph classification/regression baseline would then apply.**
>
> The reviewer makes an **excellent, insightful observation** about PLATO’s first MLP layer! PLATO’s problem setting can indeed be reformulated as graph classification/regression. However, **a graph classification/regression formulation does not work for three key reasons.**
> 1. *Every sample would correspond to a subgraph that is structurally equivalent.* Every sample’s subgraph would have the exact same nodes and edges while only the node attributes vary. However, graph classification/regression models assume that different samples correspond to structurally distinct subgraphs [68, 26, 25].
> 2. *Every subgraph would be too large for typical graph classification/regression approaches.* For example, the smallest subgraph in PLATO’s datasets would have 12,639 nodes. However, the largest subgraph in the Open Graph Benchmark’s (OGB) graph classification challenge has only 244 nodes [25].
> 3. *There would be too few samples (e.g. subgraphs) to learn an effective graph classification/regression model.* The PLATO $d \gg n$ dataset with the largest number of samples would only have 924 subgraphs. However, the smallest number of subgraphs in the Open Graph Benchmark’s graph classification task has 41,127 graphs [25].
>
> **We attempted GCN-based graph classification/regression models for the PLATO problem setting but they did not work for the reasons described above.** We thank the reviewer for their incredibly insightful comment and will add this discussion to the revised manuscript!
>
> ---
>
> > **The authors should compare to graph, tabular, and graph + tabular baselines.**
>
> Graph classification/regression baselines do not apply as described above. For tabular deep learning baselines, we already compare to methods from recent benchmarks [1, 2]. For tabular + graph baselines, we already compare to three graph regularization methods that use the KG. PLATO outperforms all baselines on all $d \gg n$ datasets (Table 1).
>
> ---
>
> > **If PLATO is a tabular deep learning method, there are two limitations: (1) PLATO is limited to a MLP,  (2) each feature has a corresponding node in the graph.**
>
> **Expanding PLATO to architectures beyond a MLP and to datasets where some input features are not in the KG is exciting future work! However, these are not fundamental limitations.** First, we do not claim PLATO is a general regularization method for tabular deep learning. Instead, PLATO is a specific architecture in which the first layer of a MLP’s weights are regularized by an auxiliary KG. Second, future work could expand PLATO to problem settings where only some input features are nodes in the KG! With a MLP in its current problem setting, PLATO already outperforms 10 state-of-the-art baselines on 6 $d \gg n$ datasets by up to +10.19%.
>
> ---
>
> > **It seems that performance gains come solely from using the auxiliary KG rather than regularization. Do any tabular baselines also leverage the KG?**
>
> **Three tabular baselines use the auxiliary KG.** GraphNet, Network-Constrained LASSO (NC LASSO), and Network LASSO are graph regularization methods that use the KG to smooth the weights in a regression model. **PLATO outperforms these baselines by > 19%** (Rebuttal Table 1, Table 1).
>
> We also do not claim that PLATO’s performance comes solely from “regularization.” We instead emphasize that PLATO’s ability to incorporate prior information is key to performance. Indeed, Table 3 shows that having more information in the KG (i.e. the full KG) leads to better performance than having less information (i.e. the feature-only KG), supporting our main claim.
>
> ---
>
> > **The paper claims to follow evaluation protocols from [1]. However, [1] used Bayesian HPO framework Optuna.**
>
> **[1] indeed uses Optuna.** However, we based PLATO on rigorous evaluation protocols in both [1] and [2], two recent tabular learning benchmarks. **[2] uses a random hyperparameter search like PLATO’s.** We reference [1] and [2] since both include hyperparameter ranges for baseline models.
>
> ---
>
> > **The authors should add tabular deep learning and transfer learning methods to related work.**
>
> We will add all suggested references, include tabular transfer learning as an alternative, and revise the statements suggested by the reviewer!
>
> ---
>
> > **Can PLATO handle categorical features?**
>
> **Yes, PLATO can handle categorical features.** If the input features are categorical rather than numerical, PLATO would simply learn different weights in the first and later layers of the MLP.
>
> ---
>
> > **Line 101 says PLATO "represents a new regularization mechanism".**
>
> **We only mean to claim that PLATO regularizes a MLP.** We do not mean to claim that PLATO can regularize any tabular model! We will remove the phrase that PLATO “represents a new regularization mechanism” in Related Work.
>
> ---
>
> > **References**
>
> [1] Gorishniy et al. “Revisiting deep learning models for tabular data.” NeurIPS (2021).
>
> [2] Grinsztajn et al. “Why do tree-based models still outperform deep learning on typical tabular data?” NeurIPS (2022).
>
> [68] Yang et al. “Do transformers really perform badly for graph representation?” NeurIPS (2021).
>
> [26] Hu et al. “Strategies for pre-training graph neural networks.” ICLR (2020).
>
> [25] Hu et al. “Open Graph Benchmark.” NeurIPS (2020).
>
> ---

---

> > ### Author Response · Authors · 2023-08-19
> > **Follow-up**
> >
> > We thank the reviewer for their time and thoughtful questions! We wanted to gently follow up before the discussion period closes. If our response has satisfied the reviewer, we would appreciate the reviewer considering raising their score. Thank you!

---

> > ### Author Response · Authors · 2023-08-21
> > **Final Follow-Up**
> >
> > We wanted to thank the reviewer one final time! We also wanted to provide a last, gentle follow up before the discussion period closes. If our response has satisfied the reviewer, we would appreciate the reviewer considering raising their score. Thank you!

---

> > > ### Comment · Reviewer_87H1 · 2023-08-21
> > > **Response to the author rebuttal**
> > >
> > > Dear Authors,
> > >
> > > Thank you for your thorough response. While I do not completely agree regarding the infeasibility of graph-based baselines because graph convolutional neural networks certainly can work on graphs with tens of thousands of nodes, and 924 samples may be enough for learning some signal, your point about the structural similarity of graphs is fair. Additionally, I still believe that the assumption of each feature having a corresponding node in the graph is limiting as well as the applicability of PLATO only to MLPs. However, thank you for your note regarding the three graph-aware tabular baselines and your clarifications regarding the categorical features. As my concerns are now partially addressed, I am raising my score.

---

### Official Review · Reviewer_E2W5 · 2023-07-07

**Soundness:** 2 fair
**Presentation:** 3 good
**Contribution:** 3 good
**Rating:** 5
**Confidence:** 3

**Summary:**

This paper introduces PLATO, an approach handles tabular deep learning challenges when the number of features (d) significantly outnumbers the available instances (n). The authors propose the integration of an auxiliary knowledge graph into the learning process to help the model better understand the relationships between features and hence make more accurate predictions. The author conducted extensive experiments to validate the effectiveness of their approach, involving scenarios with different data sparsity levels, diverse knowledge graph structures, and various types of tabular data. The results demonstrate the proposed method outperforms conventional deep learning approaches on tabular data when the number of features is much larger than the number of instances.

**Strengths:**

This paper proposed an original approach to a longstanding issue in machine learning: the 'curse of dimensionality' in situations where the feature dimension (d) greatly exceeds the number of samples (n). Through combining several existing ideas in deep learning and knowledge representation, It creates a potentially transformative technique to address challenge of high-dimensionality in tabular data.

This paper provided empirical evidence of the method's advancement. The proposed method outperforms 13 state-of-the-art baselines across 6 d >> n datasets by up to 10.19%.

The paper is generally well written, with clear explanations of the main concepts and methodology.

**Weaknesses:**

There has been existing work combining knowledeg graph with deep learning on tabular dataset, and the distinctness of proposed method from previous work is not obviously illustrated.

The robustness and scalability analysis is lacked, how does performance vary with the size and the complexity of the knowledge graph? Does this method scale to large datasets? It would be useful to include a discussion of these issues.

The proposed method PLATO doesn't show evident advancement compared to selected baselines on 4 d~n datasets, especially for XGBoost, which performance seems to keep up with PLATO and has been proposed over five years, justifying the weakness of advancement for the selected baselines.

**Questions:**

Since PLATO utilizes auxiliary knowledge graph to pretrain an embedding for each input feature, will other pretrained methods achieve same competitive performance on the selected dataset?

How diverse or domain-specific is the auxiliary knowledge graph? Can the proposed method adapt to various domains or does it require specific types of knowledge graphs?

Can the auxiliary knowledge graph combine with other advanced models rather than a simple MLP in the experiment's setting.

**Limitations:**

The reviewer does not see potential negative social impact of the work.

---

> ### Author Rebuttal · Authors · 2023-08-10
>
>
> > **Overall response**
>
> We **thank** the reviewer! The reviewer highlights our “**original approach to** a longstanding issue in machine learning: **the ‘curse of dimensionality’.”** The reviewer says PLATO is a “**potentially transformative technique** to address [the] challenge of high-dimensionality in tabular data” and emphasizes the “empirical evidence of the method’s advancement.” Finally, the reviewer has helped us clarify PLATO’s differentiation from prior work combining tabular data with knowledge graphs (KGs). **While prior work integrating tabular data with a KG models input samples as nodes in the KG, PLATO models input features as nodes in the KG.**
>
> ---
>
> > **How is PLATO different from prior work that combines a knowledge graph (KG) with deep learning on a tabular dataset?**
>
> **Prior work models input samples as nodes in the auxiliary KG while PLATO models input features as nodes in the auxiliary KG.** For traditional prediction problems combining an auxiliary KG with tabular data, each input sample is a node in the KG (i.e. users in a social network) and each node has a feature vector from the tabular data (i.e. a vector of the age, height, and weight of the user) [1]. However, PLATO operates in a distinct problem setting. In PLATO, every input feature (i.e. the column name) is a node in the KG. The KG, containing prior relationships between input features, is then used to regularize a separate MLP which predicts the label for each input sample.
>
> For example, consider a tabular dataset in which each row is a cancer patient, each column is a gene, and each value is the level of each gene in a patient’s tumor. In PLATO, every gene (e.g. gene name) is a node in an auxiliary KG. The auxiliary KG is then used to regularize a MLP which, for each patient, takes in the vector of gene levels and outputs the patient’s survival. Knowledge of prior relationships between genes (e.g. input features) enables strong performance even when there are far more genes (e.g. tens-of-thousands) than patients (e.g. hundreds) in the tabular dataset.
>
> [1] Thomas et al. "Graph neural networks designed for different graph types: A survey." TMLR (2022).
>
> ---
>
> > **How does PLATO’s performance vary with the size of the KG?**
>
> We conduct an **ablation study to assess how PLATO’s performance varies with the size of the KG (e.g. number of edges)**. We randomly remove edges from the KG and measure PLATO’s performance on the BRCA dataset. **With only 50% of the KG’s edges, PLATO still has 71% of the performance as PLATO with 100% of the KG’s edges** (0.412 vs 0.583). Generally, PLATO’s performance improves as the KG grows and there are fewer missing edges (e.g. less incomplete information) (Rebuttal Table 2).
>
> ---
>
> > **Does PLATO scale to large datasets?**
>
> PLATO’s key benefit is the ability to learn from datasets with many features but few samples (e.g. $d \gg n$). **PLATO thus scales to datasets that are large in the number of features** but small in the number of samples.
>
> ---
>
> > **For datasets where the number of features is comparable to the number of samples, PLATO exhibits similar performance to XGBoost.**
>
> **XGBoost is currently state-of-the-art for tabular datasets with more samples than features** (e.g. $n \gg d$) [1, 2]. We thus consider it a **strength**, not a weakness, **that PLATO matches XGBoost when the number of samples is comparable to the number of features** (e.g. $n \sim d$). The key result, however, is that **PLATO outperforms XGBoost when the number of samples is far smaller than the number of features** (e.g. $d \gg n$) (Table 1). In other words, PLATO continues to exhibit strong performance in lower-sample regimes where XGBoost’s relative performance falls substantially.
>
> [1] Gorishniy et al. “Revisiting deep learning models for tabular data.” NeurIPS (2021).
>
> [2] Grinsztajn et al. “Why do tree-based models still outperform deep learning on typical tabular data?” NeurIPS (2022).
>
> ---
>
> > **PLATO uses the auxiliary KG to pretrain an embedding for each input feature. Will other pretrained methods achieve the same performance as PLATO?**
>
> **We are not familiar with other methods that can use PLATO’s pre-trained input feature embeddings or other pre-trained methods that apply.** First, other methods cannot use PLATO’s pre-trained feature embeddings because of PLATO’s unique problem setting. In PLATO, every input feature in the tabular dataset (e.g. column name) is a node in the auxiliary KG while in other approaches, every input sample (e.g. row name) is a node. PLATO’s input feature embeddings thus do not apply to other models. In fact, how to use these input feature embeddings is nontrivial, and PLATO’s use of them to infer the weights in the first layer of a MLP is itself an innovation. Second, alternative pre-training techniques don’t apply to PLATO’s problem setting because the information that PLATO uses in pre-training is an auxiliary KG rather than another tabular dataset.
>
> **Nonetheless, we still ensure that PLATO outperforms baselines which use the auxiliary KG more generally.** For example, we compare PLATO to three graph regularization methods that use the feature relationships in the KG to smooth the weights in a regression model. PLATO outperforms graph regularization baselines across all $d \gg n$ datasets by >19% (Rebuttal Table 1, Table 1).
>
> ---
>
> > **How diverse or domain-specific is the auxiliary KG? Can the proposed method adapt to various domains or does it require specific types of KG?**
>
> **PLATO can be applied to various domains and does not assume whether the KG used is domain-specific or general.** As long as the KG contains each input feature from the tabular data as a node and also contains prior information about the input features, PLATO can be applied.
>
> ---
>
> > **PLATO’s framework applies the auxiliary KG to a MLP. Can PLATO apply the auxiliary KG to more advanced models?**
>
> Expanding PLATO beyond a MLP is an **exciting path for future work**!
>
> ---

---

> > ### Author Response · Authors · 2023-08-19
> > **Follow-up**
> >
> > We thank the reviewer for their time and thoughtful questions! We wanted to gently follow up before the discussion period closes. If our response has satisfied the reviewer, we would appreciate the reviewer considering raising their score. Thank you!

---

> > > ### Comment · Reviewer_E2W5 · 2023-08-21
> > >
> > > Thank you for the response. I have no further questions.

---

### Author Rebuttal · Authors · 2023-08-10

> **Overall response**


We are extremely grateful for the highly detailed and constructive feedback on our manuscript! The reviewers describe PLATO as a **“potentially transformative technique to address [the] challenge of high-dimensionality in tabular data”** (E2W5), say that **“the mechanism to induce KG regularization is novel”** (p8DV), emphasize that PLATO “could be **very valuable to numerous scientific fields**” (p8DV) including an “**abundance of prediction problems in the biomedical space**” (dZQm), and say that “the approach is non-obvious and effective, while also being intuitive and clearly explained” (dZQm).


PLATO is a deep learning method for tabular datasets with orders of magnitude more features than samples. PLATO tackles a **unique problem setting in which every input feature (e.g. column name) in a tabular dataset is a node in an auxiliary knowledge graph (KG)**. By contrast, prior work combining tabular data with KGs models every input sample (e.g. row name) as a node. Since PLATO leverages prior information about the input features, PLATO can achieve strong predictive performance, even when there are orders of magnitude more input features than samples (e.g. $d \gg n$). On **6 datasets with $d \gg n$**, we demonstrate that **PLATO outperforms 10 state-of-the-art baselines by up to +10.19%.**

In our rebuttal, we address the comments of the review team. This has led to multiple improvements including: illustrating that PLATO outperforms baselines that also use an auxiliary KG (E2W5, 87H1, p8DV), clarifying PLATO’s contributions (p8DV, 87H1), explaining PLATO’s limitations (E2W5, 87H1, dZQm), and enhancing the related work (87H1). We answer all reviewer comments below and thank them for the time they put into reading, understanding, and evaluating our paper! Tables for the rebuttal (e.g. Rebuttal Table 1, Rebuttal Table 2) are in the attached PDF.

---

---

### Decision · Program_Chairs · 2023-09-21

**Decision:**

Accept (poster)

**Comment:**

This paper introduces a creative and elegant way to use prior knowledge when training NN on tabular data. The idea is to create a graph such that a feature in the data corresponds with a vertex in the graph. The graph is used in the training process to create a model. The idea is very appealing, and the empirical results are promising. However, the specific implementation can be improved by considering it as a graph learning problem and using GNNs as proposed by one of the reviewers.